# RETRIEVAL MEETS LONG CONTEXT LARGE LANGUAGE MODELS

**Peng Xu[†], Wei Ping[†], Xianchao Wu, Lawrence McAfee**
**Chen Zhu, Zihan Liu, Sandeep Subramanian, Evelina Bakhturina**
**Mohammad Shoeybi, Bryan Catanzaro**
NVIDIA
[†]{pengx, wping}@nvidia.com

## ABSTRACT

Extending the context window of large language models (LLMs) is getting popular recently, while the solution of augmenting LLMs with retrieval has existed for years. The natural questions are: *i) Retrieval-augmentation versus long context window, which one is better for downstream tasks? ii) Can both methods be combined to get the best of both worlds?* In this work, we answer these questions by studying both solutions using two state-of-the-art pretrained LLMs, i.e., a proprietary 43B GPT and Llama2-70B. Perhaps surprisingly, we find that LLM with 4K context window using simple retrieval-augmentation at generation can achieve comparable performance to finetuned LLM with 16K context window via *positional interpolation* on long context tasks, while taking much less computation. More importantly, we demonstrate that retrieval can significantly improve the performance of LLMs regardless of their extended context window sizes. Our best model, retrieval-augmented Llama2-70B with 32K context window, outperforms GPT-3.5-turbo-16k and Davinci003 in terms of average score on nine long context tasks including question answering, query-based summarization, and in-context few-shot learning tasks. It also outperforms its non-retrieval Llama2-70B-32k baseline by a margin, while being much faster at generation. Our study provides general insights on the choice of retrieval-augmentation versus long context extension of LLM for practitioners.

## 1 INTRODUCTION

The long context large language models (LLM) have recently received a lot of attention in production (e.g., Anthropic, 2023; OpenAI, 2023b), research community (e.g., Chen et al., 2023; Liu et al., 2023; Tworkowski et al., 2023), and open source community (e.g., Kaiokendev, 2023). Although the *approximate* attention methods have been studied for years (e.g., Tay et al., 2022a) (due to the quadratic time and memory complexities of self-attention mechanism in sequence length), the recent advance for long context LLMs with *exact* attention is mainly driven by the development of faster GPU with more memory and memory-efficient exact attention (Dao et al., 2022; Dao, 2023).

An alternative and long-standing solution for handling long context is *retrieval*. Specifically, the LLMs only read relevant context retrieved from a standalone retriever (e.g., Karpukhin et al., 2020; Wang et al., 2022; Lin et al., 2023), which is much easier to scale [1] and runs orders of magnitudes faster than LLMs for selecting relevant context. Conceptually, the retrieval-augmented decoder-only LLM can be viewed as applying the sparse attention over its long context window, where the sparsity pattern is not predefined as Child et al. (2019) but determined by the standalone retriever. In other words, unretrieved context is treated as irrelevant and has zero-valued attention weights.

Given the surge of interest in long context LLM research and much more required computation at inference [2], it is still unclear for practitioners whether extending the context window of LLM

---

[1]The dense embedding retriever can easily retrieve context from billions of tokens using the fast similarity search library (Johnson et al., 2019).

[2]For example, the price of GPT-4 with 32k context length is twice the 8k context model.

provides higher accuracy than the retrieval-augmentation for downstream tasks with informative queries. Moreover, it would be compelling if we could combine the strength of both methods and achieve even higher accuracies. In this work, we attempt to answer the above questions through a comprehensive study.

Specifically, we make the following contributions:

1. We perform comprehensive study using two state-of-the-art LLMs, a proprietary 43B pre-trained GPT and Llama2-70B (Touvron et al., 2023b) on 9 downstream long context tasks, including single and multi document question answering (QA), query-based summarization, and in context few-shot learning tasks.

2. We demonstrate that retrieval-augmentation significantly improves the performance of 4K context LLMs. Perhaps surprisingly, we find this simple retrieval-augmented baseline can perform comparable to 16K long context LLMs, i.e., average score 29.32 vs. 29.45 by using GPT-43B, and 36.02 vs. 36.78 by using Llama2-70B, while using much less computation.

3. Furthermore, we demonstrate that the performance of long context LLM (i.e., 16K or 32K) can still be improved by retrieval, especially for the larger Llama2-70B. As a result, our best model, retrieval augmented Llama2-70B-32k-ret with 32K context window (avg. score 43.6), outperforms GPT-3.5-turbo-16k (avg. score 42.8) and Davinci-003 in terms of average score. It also largely outperforms its non-retrieval Llama2-70B-32k baseline (avg. score 40.9), while can be much faster at generation (e.g., 4× faster on NarrativeQA).

We organize the rest of the paper as follows. We discuss related work in Section 2, and present the experimental setup in Section 3. We report results in Section 4 and conclude the paper in Section 5.

## 2 RELATED WORK

In this section, we discuss the related work in long context LLM, efficient attention methods, and retrieval-augmented language models.

### 2.1 PARALLEL WORK

When we are preparing this manuscript, we notice that a concurrent work (Bai et al., 2023) (arXived on 28 Aug 2023) also studies the impact of retrieval on long context LLM, including black-box model GPT-3.5-Turbo-16k (OpenAI, 2022), white-box model Llama2-7B-chat-4k (Touvron et al., 2023b), and ChatGLM2-6B-32k (Zeng et al., 2022). Different from our findings, they find that retrieval is only helpful for Llama2-7B-chat-4k with 4K context window, but not helpful for long context model, i.e., GPT-3.5-Turbo-16k and ChatGLM2-6B-32k. We hypothesize the major reasons are: *i)* it is challenging to do controlled experiments using black-box APIs, *ii)* the white-box LLMs used in their study are relatively small, thus they have limited zero-shot capability of incorporating context through retrieval. Our conclusions are drawn from much larger LLMs. In particular, our best long context model Llama2-70B-32k performs as well as Davinci003 and GPT-3.5-turbo-16k, while it can still be further enhanced by retrieval (see Table 3).

### 2.2 LONG CONTEXT LARGE LANGUAGE MODELS

Over the past few years, pretraining large language models (LLMs) with long context window becomes a viable solution thanks to faster GPU with more memory and memory-efficient exact attention (e.g., Dao et al., 2022). For example, the context window for pretrained LLM have been increased from 1024 of GPT-2 (Radford et al., 2019), 2048 of GPT-3 (Brown et al., 2020), 4096 of Llama 2 (Touvron et al., 2023b), to 8192 of GPT-4 (OpenAI, 2023a). However, further extending the context window in pretraining can be challenging, because, *i)* pretraining LLM from scratch with long context (e.g., >16K tokens) is very expensive due to the quadratic time and memory complexities of exact attention, and *ii)* most of documents in pretraining corpus (e.g., Common Crawl) are relatively short.

Most recently, researchers start to extend the context window of LLMs with continued training or fine-tuning (e.g., Kaiokendev, 2023; Nijkamp et al., 2023; Chen et al., 2023; Tworkowski et al., 2023; Mohtashami & Jaggi, 2023). Tworkowski et al. (2023) introduced LongLLaMA by fine-tuning the

3B and 7B OpenLLaMA checkpoints with contrastive training on 8K context length. Landmark attention (Mohtashami & Jaggi, 2023) extends the context length of LLaMA 7B from 4K to 32K by introducing "landmark tokens" to represent blocks of the context and fine-tuning the attention to use landmark tokens for selecting relevant blocks. Chen et al. (2023) and Kaiokendev (2023) introduced *positional interpolation* to extend the context window sizes of RoPE-based (Su et al., 2021) pretrained LLMs. In particular, Chen et al. (2023) demonstrates promising results on LLaMA 7B to 65B (Touvron et al., 2023a) with minimal fine-tuning effort (within 1000 steps). ALiBi (Press et al., 2021) extrapolates context window length by removing the positional embeddings while simply biasing the key-query attention scores with a linear penalty that is proportional to their distance, so one does not need finetuning for context window extrapolation. Ratner et al. (2023) chunks long context into multiple sub-windows and re-use the positional embeddings across these windows, thus can handle longer context without any further finetuning. In this work, we apply *positional interpolation* method to extend the 4K context window of a proprietary 43B pretrained LLM and Llama2-70B (Touvron et al., 2023b) to 16K and 32K, as they both use rotary position embedding at pretraining. In terms of evaluation, we focus on downstream task performance (e.g., Shaham et al., 2023; Bai et al., 2023) after instruction tuning (Wei et al., 2021).

There are other studies showing the interplay between retrieval-augmentation and long context LLM. Liu et al. (2023) performs the black-box evaluation for the long context capability of existing LLM products, including ChatGPT 3.5 (OpenAI, 2022), GPT-4 (OpenAI, 2023a), Claude (Anthropic, 2023), in retrieval-augmented setting, and identify the "lost in the middle" phenomenon in these models.

## 2.3 EFFICIENT ATTENTION METHODS

In previous study, many approximate attention methods (Tay et al., 2022a) have been introduced for dealing with the quadratic complexity of self-attention that becomes a computational bottleneck for long context. They can be grouped into the following categories: *i)* Sparse attention mechanisms with predefined sparsity patterns (e.g., Child et al., 2019; Parmar et al., 2018; Ho et al., 2019; Beltagy et al., 2020; Zaheer et al., 2020; Zhu et al., 2021), *ii)* recurrence-based method (Dai et al., 2019; Bulatov et al., 2022), *iii)* low-rank projection attention (e.g., Wang et al., 2020; Xiong et al., 2021; Tay et al., 2021; Zhu et al., 2021), *iv)* memory-based mechanisms (e.g., Rae et al., 2020; Liu et al., 2018), *v)* similarity and clustering based methods (e.g., Kitaev et al., 2020; Tay et al., 2020; Roy et al., 2021). These approximate methods introduce inductive bias (e.g., predefined sparsity) that can fit well for specific domain, but may reduce model quality in general LLM training.

Most recently, FlashAttention (Dao et al., 2022; Dao, 2023) is introduced to speed up the exact attention computation by accounting for reads and writes between levels of GPU memory. FlashAttention is particularly useful for handling longer sequences.

## 2.4 RETRIEVAL-AUGMENTED LANGUAGE MODELS

Retrieval has been integrated into language models for years to improve perplexity (Borgeaud et al., 2022; Wang et al., 2023), factual accuracy (Nakano et al., 2021), downstream task accuracy (Guu et al., 2020; Izacard & Grave, 2021; Izacard et al., 2022; Lewis et al., 2020), and in-context learning capability (Huang et al., 2023). Combined with a standalone retriever (Karpukhin et al., 2020; Wang et al., 2022; Lin et al., 2023), retrieval-augmented LLM is well established for handling question answering with long document and in open-domain. In previous study, language models have been augmented with retrieval at inference (Khandelwal et al., 2019; Yogatama et al., 2021), fine-tuning (Izacard et al., 2022; Lewis et al., 2020; Guu et al., 2020), and pretraining (Borgeaud et al., 2022; Izacard et al., 2022; Wang et al., 2023). There are also methods that try to integrate LLM and retriever in a single model and build the end-to-end solution (e.g., Jiang et al., 2022; Shi et al., 2023). However, most of previous works mainly study retrieval-augmentation for LLMs that have around 10 billion parameters, except a few recent ones (e.g., Shi et al., 2023).

In this work, we focus on decoder-only LLMs with 43B and 70B parameters trained on trillions of tokens, because the LLMs at such scale exhibit strong zero-shot capability to incorporate context after instruction tuning (Wei et al., 2021; 2022).

## 3 EXPERIMENTAL SETUP

In this section, we present the details of our experimental setup.

### 3.1 LARGE LANGUAGE MODELS

We focus on comparing the zero-shot capability of integrating long context information for generative QA or summarization tasks via retrieval or LLM's own self-attention mechanism. In contrast to most existing works that focus on relatively small models (e.g., 3B or 7B) (Kaiokendev, 2023; Nijkamp et al., 2023; Tworkowski et al., 2023; Mohtashami & Jaggi, 2023), we gather the insights by exploring model sizes that are larger than 40B after instruction tuning, as previous study suggests that instruction tuning becomes effective when the decoder-only LLM has around 50B parameters (Wei et al., 2021; 2022).

Specifically, we experimented with two pretrained GPT models, a proprietary Nemo GPT-43B and Llama2-70B. GPT-43B is a 43 billion parameter model that is trained with 1.1T tokens with 70% English corpus and the other 30% for multilingual and code data. For the English pretraining corpus, GPT-43B used Common Crawl web archive (WARC), Wikipedia, Reddit, Books, Gutenberg, ArXiv, StackExchange, PubMed, etc. It contains 48 layers with the hidden dimension of 8,192. It is trained with a sequence length of 4,096 and RoPE embeddings (Su et al., 2021). The other Llama2-70B is a public available 70B GPT model trained on 2T tokens using around 90% English data. It contains 80 layers with the hidden dimension of 8,192. It also has the context window size of 4,096 and trained with RoPE embeddings.

### 3.2 DATASETS AND METRICS

In this study, we include seven datasets ranging from single document QA, multi document QA, to query-based summarization for our zero shot evaluations. Specifically, we include four datasets from the validation set of the Scroll benchmark (Shaham et al., 2022).

- **QMSum (QM)** (Zhong et al., 2021) is a query-based summarization dataset, consisting of meetings' transcripts and their corresponding summaries from multiple domains such as academic, industrial product. In this task, a meeting dialogue transcript is given, and a question to summarize certain topic is raised about the dialogue, such as "what is agreed between them". The answer generally contains a few sentences.

- **Qasper (QASP)** (Dasigi et al., 2021) is a question answering dataset over NLP papers filtered from the Semantic Scholar Open Research Corpus (S2ORC) (Lo et al., 2020). Qasper contains abstractive, extractive, and yes/no questions, as well as unanswerable ones. In this task, one script is provided together with an information seeking question, such as "which multilingual approaches do they compare with?". A model needs to give a short answer by reasoning over the given context.

- **NarrativeQA (NQA)** (Kočiský et al., 2018) is an established question answering dataset over entire books from Project Gutenberg[3] and movie scripts from a list of websites. In this task, the given passage is transcribed from books and is usually noisy. A model is required to generate a short phrase by reasoning over the long and noisy text.

- **QuALITY (QLTY)** (Pang et al., 2022) is a question answering dataset over stories and articles collected from several resources, such as Project Gutenberg and the Open American National Corpus[4]. Different from all the other tasks, this is a multi-choices dataset and a model is required to select one among four given choices.

We take another three datasets from LongBench (Bai et al., 2023).

- **HotpotQA (HQA)** (Yang et al., 2018) is a Wikipedia-based question-answer dataset. Different from above single hot datasets, HQA is a multi-hop dataset where multiple supporting documents are required to be read for answering and reasoning and the questions are diverse and not constrained to any pre-existing knowledge bases.

---

[3] https://www.gutenberg.org/
[4] https://anc.org/

| | QM | QASP | NQA | QLTY | MSQ | HQA | MFQA |
|---|---|---|---|---|---|---|---|
| # of samples | 200 | 1,726 | 2,000 | 2,000 | 200 | 200 | 150 |
| avg doc length | 14,140 | 4,912 | 84,770 | 6,592 | 16,198 | 13,319 | 7,185 |
| avg top-5 chunks | 2,066 | 2,071 | 2,549 | 2,172 | 2,352 | 2,322 | 2,385 |
| avg top-10 chunks | 4,137 | 3,716 | 5,125 | 4,018 | 4,644 | 4,554 | 4,305 |
| avg top-20 chunks | 8,160 | 4,658 | 10,251 | 5,890 | 9,133 | 8,635 | 6,570 |

Table 1: Statistics of seven datasets used for zero-shot evaluation. All lengths are counted by the number of tokens using Llama2-70B tokenizer, and "avg top N chunks" denotes the average number of tokens from the top N retrieved chunks. Figure 2 gives more details.

- **MuSiQue (MSQ)** (Trivedi et al., 2022) is another multi-hop question answering dataset. Compared to HQA, MSQ requires connected reasoning by reducing potential reasoning shortcuts, minimizing train-test leakage, and including harder distractor contexts. Thus, MSQ is much harder task than HQA and significantly less cheatable.
- **MultiFieldQA-en (MFQA)** (Bai et al., 2023) was manually curated to better test the model's long context understanding ability across diverse fields. The evidences from multiple sources, including legal documents, government reports, encyclopedias, and academic papers, are fairly randomly located in the documents to avoid biases that might occur at the beginning or ending of the documents.

The full details of the dataset can be found in Table 1. We can see that our evaluation datasets have a wide range of average document length from 4.9k (QASP) to 84k (NQA). Therefore, for the baseline model without retrieval, we truncate the document accordingly to fit into the input sequence length.

Following the official metrics, we report the geometric mean of ROUGE scores (i.e., ROUGE-1/2/L) (Lin, 2004) for QM, the exact matching (EM) score for QLTY, and F1 scores for the remaining five datasets QASP, NQA, MSQ, HQA and MFQA.

### 3.3 CONTEXT WINDOW EXTENSION

We extend the context window length with position interpolation method (Chen et al., 2023), as it is simple and effective for RoPE embeddings. We extend the 4K context window to 16K for GPT-43B. For Llama2, we extend its 4K context window to 32k for Llama2-7B and both 16K and 32K for Llama2-70B. We follow Chen et al. (2023) and finetune both LLMs on the Pile dataset (Gao et al., 2021) with batch size as 128, constant learning rate of 5e-6 to adapt the position embeddings.

### 3.4 RETRIEVAL

For the retriever, we experimented with three retrievers: 1) *Dragon* (Lin et al., 2023) as it achieves state-of-the-art results on both supervised and zero-shot information retrieval benchmarks (Thakur et al., 2021). Dragon is a dual encoder model that consists of a query encoder and a context encoder. 2) a widely used *Contriever* model (Izacard et al., 2021). Following the MoCo technique (He et al., 2020), Contriever used a simple contrastive learning framework to pre-train models for information retrieval. It was trained without supervision and achieved competitive results with BM25 for R@100 on the BEIR benchmark (Thakur et al., 2021), and 3) *OpenAI embedding*[5]. For the OpenAI embedding model, we use the latest "text-embedding-ada-002" as recommended by OpenAI. It accepts 8,191 maximum input tokens for one sequence with an output vector of 1,536 dimensions. The cosine similarities are then computed between the questions and the list of contexts for retrieval ranking.

To use these retrievers, we first chunk each context document with 300 words, and then we encode both the questions and all chunks independently with corresponding encoders. The most relevant N chunks, ranked by the dot product of the question embedding and chunk embedding, are then concatenated together (following the left to right order from the most relevant to least relevant) as the context of the prompt for generation. Table 1 shows the statistics of the top N retrieved chunks while Figure 2 in the Appendix gives more details of the token length distribution of all seven datasets. We

---

[5]https://platform.openai.com/docs/guides/embeddings

| Model | Seq len. | Avg. | QM | QASP | NQA | QLTY | MSQ | HQA | MFQA |
|---|---|---|---|---|---|---|---|---|---|
| GPT-43B | 4k | 26.44 | 15.56 | 23.66 | 15.64 | 49.35 | 11.08 | 28.91 | 40.90 |
| + ret | 4k | 29.32 | 16.60 | 23.45 | 19.81 | 51.55 | 14.95 | 34.26 | 44.63 |
| GPT-43B | 16k | 29.45 | 16.09 | 25.75 | 16.94 | 50.05 | 14.74 | 37.48 | 45.08 |
| + ret | 16k | **29.65** | 15.69 | 23.82 | 21.11 | 47.90 | 15.52 | 36.14 | 47.39 |
| Llama2-70B | 4k | 31.61 | 16.34 | 27.70 | 19.07 | 63.55 | 15.40 | 34.64 | 44.55 |
| + ret | 4k | 36.02 | 17.41 | 28.74 | 23.41 | 70.15 | 21.39 | 42.06 | 48.96 |
| Llama2-70B | 16k | 36.78 | 16.72 | 30.92 | 22.32 | **76.10** | 18.78 | 43.97 | 48.63 |
| + ret | 16k | 37.23 | **18.70** | 29.54 | 23.12 | 70.90 | 23.28 | 44.81 | 50.24 |
| Llama2-70B | 32k | 37.36 | 15.37 | **31.88** | 23.59 | 73.80 | 19.07 | 49.49 | 48.35 |
| + ret | 32k | **39.60** | 18.34 | 31.27 | **24.53** | 69.55 | **26.72** | **53.89** | **52.91** |
| Llama2-7B | 4k | 22.65 | 14.25 | 22.07 | 14.38 | 40.90 | 8.66 | 23.13 | 35.20 |
| + ret | 4k | **26.04** | 16.45 | 22.97 | 18.18 | 43.25 | 14.68 | 26.62 | 40.10 |
| Llama2-7B | 32k | **28.20** | 16.09 | 23.66 | 19.07 | 44.50 | 15.74 | 31.63 | 46.71 |
| + ret | 32k | 27.63 | 17.11 | 23.25 | 19.12 | 43.70 | 15.67 | 29.55 | 45.03 |

Table 2: Comparison of model variants (GPT-43B, Llama2-7B, Llama2-70B) with sequence length ranging from 4k to 32k under seven datasets. "ret" denotes using the best retriever (Dragon or Contriever or OpenAI embeddings) and here we used top-5 for the retriever.

can see that top-5 chunks can all fit into 4k sequence length (except few outliers) while top-10 and top-20 chunks can fit into 16k sequence length.

## 3.5 INSTRUCTION TUNING

To train the pretrained LLMs to follow instructions for question answering or text summarization, we also performed instruction tuning. We first construct a blend of instruction tuning datasets consisting of 102K training samples from the Soda dataset (Kim et al., 2022), ELI5 dataset (Fan et al., 2019), FLAN dataset (Wei et al., 2021) , Open Assistatant dataset (Köpf et al., 2023), Dolly (Conover et al., 2023) and a proprietary sourced conversational dataset, to adapt all foundation models to follow instructions. In terms of the template, we use "System: {System}\n\nUser: {Question}\n\nAssistant: {Answer}" as the format to support multi-turn dialogue training. As all of the tasks contain the context information for reasoning over at inference time, we add the context before the dialogue, i.e. "System: {System}\n\n{Context}\n\nUser: {Question}\n\nAssistant: {Answer}".

We finetune the LLM by taking the loss only on the {Answer} part with batch size 128 and learning rate of 5e-6 for 1000 steps. For the rest of the paper, results are all reported using the instruction tuned chat model on top of the foundational GPT-43B, Llama2-7B, and Llama2-70B.

## 4 RESULTS

In this section, we report the results and provide detailed analysis.

## 4.1 MAIN RESULTS

In Table 2, we compare different model variants with context lengths ranging from 4K to as long as 32K using GPT-43B and Llama2-70B. First, we find that baseline models without retrieval of 4k sequence length achieve the worst results for both GPT-43B and Llama2-70B. This is because the minimum average sequence length of all seven tasks exceeds 4096, the context window of the foundation models and therefore valuable texts get truncated randomly. As a result, retrieval is especially helpful for 4K LLMs e.g., Llama2-70B-4K is improved from 31.61 to 35.73 while GPT-43B-4K is improved from 26.44 to 29.32. Second, we observe that HotpotQA (HQA) especially favors long sequence models as the score improves from 34.64 to 43.97 for Llama2-70B and from 28.91 to 37.48 for GPT-43B when the sequence length increases from 4k to 16k. This is because Hotpot QA is a multi-hop dataset where the questions are not hard to answer but all intermediate hops are necessary to get correct answer. Therefore, long context are beneficial to increase the recall of incorporating all intermediate hops.

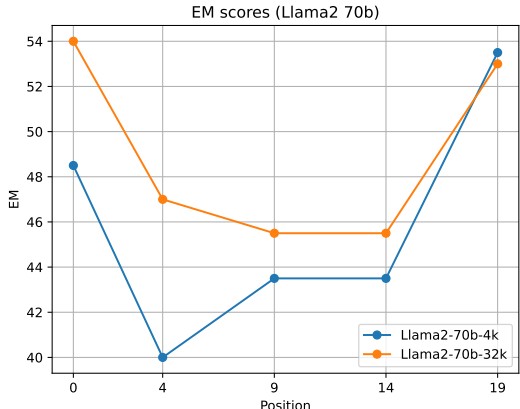

Figure 1: Llama2-70B also displays lost-in-the-middle phenomenon

| Model | Avg-7 | Avg-4* | QM* | QASP* | NQA* | QLTY* | MSQ | HQA | MFQA |
|---|---|---|---|---|---|---|---|---|---|
| Davinci003 (175B) | 39.2 | 40.8* | 16.9* | 52.7* | 24.6* | 69.0* | 22.1 | 41.2 | 47.8 |
| GPT-3.5-turbo (4k) | 38.4 | 39.2* | 15.6* | 49.3* | 25.1* | 66.6* | 21.2 | 40.9 | 49.2 |
| +ret | | | | | | | 24.4 | 49.5 | 49.5 |
| GPT-3.5-turbo-16k | 42.8 | 42.4 | 17.6 | 50.5 | 28.8 | 72.6 | 26.9 | 51.6 | 52.3 |
| +ret | | | | | | | 30.4 | 46.6 | 52.8 |
| Llama2-70B-32k | 40.9 | 42.4 | 15.6 | 45.9 | 28.4 | 79.6 | 19.1 | 49.5 | 48.4 |
| Llama2-70B-32k-ret | **43.6** | **43.0** | 18.5 | 46.3 | 31.5 | 75.6 | 26.7 | 53.9 | 52.9 |

Table 3: Comparison of our best retrieval-augmented Llama2-70B-32k-ret with GPT-3.5-turbo-16k and Davinci-003 (175B parameters). For QMSum (QM), Qasper (QASP), NarrativeQA (NQA), QuALITY (QLTY), we used the test set from the ZeroSCROLLS leaderboard as the organizers have prepared the scores of GPT-3.5-turbo (4k) and Davinci-003 (highlighted with *). Avg-7 refers to the average score of all 7 datasets, and Avg-4* refers to the average of 4 datasets from ZeroSCROLLS.

It is quite interesting that the retrieval-augmented long context LLM (e.g., 16K and 32K) can obtain better results than retrieval-augmented 4K context LLM, even they are feed with the same top 5 chunks of evidence. We hypothesize this interesting observation is related to the "lost in the middle" phenomenon (Liu et al., 2023), where the LLMs has such "U-shaped" performance curve. Specifically, LLMs are better at utilizing relevant information that occurs at the beginning or end of its input context window. To further verify the hypothesis, we conduct the "lost-in-the-middle" study following Liu et al. (2023) for Llama2-70B-4k and Llama2-70B-32k. As show in Figure 1, we confirm that the phenomenon also exists in Llama2-70B with different context lengths. In particular, the comparison of the curves from Llama2-70B-4k and Llama2-70B-32k suggests that the long context model has better accuracy for incorporating top-5 retrieved context.

Note that, we have very different observation from the conclusion drawn from LongBench work (Bai et al., 2023): *"Retrieval brings improvement for model with weak ability on long contexts, but the performance still lags behind models that have strong long context understanding capability"*. Here, we demonstrate retrieval can significantly improve the performance of both GPT-43B and Llama2-70B regardless their context window size. For example, our best retrieval-augmented Llama2-70B-32k-ret outperforms its baseline w/o retrieval by a margin, i.e., 39.60 vs. 37.36. We think the major reason for such different conclusion is that Bai et al. (2023) uses much smaller LLM with 6B and 7B parameters, which usually has relatively worse zero-shot capability to incorporate the retrieved chunked context. To further validate the hypothesis, we also report the results using Llama2-7B in Table 5. One can actually draw similar conclusions to Bai et al. (2023) . We think the underlying reasons are: i) For Llama2-7B-chat-4k, its short context length is the bottleneck for long context tasks. Thus, retrieval-augmentation largely improves the results. ii) For Llama2-7B-chat-32 and ChatGLM2-6B-32k, the context length bottleneck has been mostly removed. However, their retrieval-augmented models have limited zero-shot capability of incorporating retrieved chunks of context, due to the smaller size. As a result, retrieval is not helpful for both Llama2-7B-32k and ChatGLM2-6B-32k, which is different from large LLMs like Llama2-70B-32k in our case.

| Seq len | Setting | Avg. | QM | QASP | NQA | QLTY | MSQ | HQA | MFQA |
|---------|---------|------|-----|------|-----|------|-----|-----|------|
| 4k | baseline (w/o ret) | 31.61 | 16.34 | 27.70 | 19.07 | 63.55 | 15.40 | 34.64 | 44.55 |
| | Dragon | 35.73 | 18.14 | 29.20 | 23.39 | 70.30 | 20.09 | 41.54 | 47.45 |
| | Contriever | **36.02** | 17.41 | 28.74 | 23.41 | 70.15 | 21.39 | 42.06 | 48.96 |
| | OpenAI-embedding | 35.79 | 17.76 | 28.85 | 23.57 | 70.70 | 19.92 | 41.76 | 47.99 |
| 32k | baseline (w/o ret) | 37.36 | 15.37 | 31.88 | 23.59 | 73.80 | 19.07 | 49.49 | 48.35 |
| | Dragon | **39.60** | 18.34 | 31.27 | 24.53 | 69.55 | 26.72 | 53.89 | 52.91 |
| | Contriever | 38.85 | 17.60 | 31.56 | 23.88 | 69.00 | 26.61 | 49.65 | 53.66 |
| | OpenAI-embedding | 39.34 | 18.24 | 32.07 | 24.36 | 69.45 | 24.90 | 51.64 | 54.75 |

Table 4: Comparisons of adding top 5 retrieved chunks from different retrievers to the context under Llama2-70B.

| Seq len | Setting | Avg. | QM | QASP | NQA | QLTY | MSQ | HQA | MFQA |
|---------|---------|------|-----|------|-----|------|-----|-----|------|
| 4k | base | 31.61 | 16.34 | 27.70 | 19.07 | 63.55 | 15.40 | 34.64 | 44.55 |
| | top-5 | **35.73** | 18.14 | 29.20 | 23.39 | 70.30 | 20.09 | 41.54 | 47.45 |
| | top-10 | 34.62 | 16.54 | 28.67 | 24.38 | 68.70 | 19.00 | 42.18 | 42.84 |
| | top-20 | 34.61 | 16.52 | 28.67 | 24.38 | 68.70 | 19.00 | 42.18 | 42.84 |
| 16k | base | 36.78 | 16.72 | 30.92 | 22.32 | 76.10 | 18.78 | 43.97 | 48.63 |
| | top-5 | 37.23 | 18.70 | 29.54 | 23.12 | 70.90 | 23.28 | 44.81 | 50.24 |
| | top-10 | **38.31** | 18.41 | 30.20 | 25.53 | 73.60 | 22.78 | 47.72 | 49.91 |
| | top-20 | 36.61 | 17.26 | 29.60 | 25.81 | 72.30 | 22.69 | 41.36 | 47.23 |
| 32k | base | 37.36 | 15.37 | 31.88 | 23.59 | 73.80 | 19.07 | 49.49 | 48.35 |
| | top-5 | **39.60** | 18.34 | 31.27 | 24.53 | 69.55 | 26.72 | 53.89 | 52.91 |
| | top-10 | 38.98 | 17.71 | 30.34 | 25.94 | 70.45 | 22.80 | 55.73 | 49.88 |
| | top-20 | 38.38 | 16.36 | 30.42 | 24.42 | 69.60 | 24.51 | 54.67 | 48.65 |

Table 5: Comparisons of adding top-5/10/20 retrieved chunks to the context under 4k, 16k, and 32k input sequence lengths using Llama2-70B. More context does not always give better results.

In contrast, the larger instruction tuned LLMs like Llama2-70B has much stronger zero-shot capability to incorporate retrieved evidence. This observation is becoming more clear when one compares the gain of retrieval-augmentation between GPT-43B and Llama2-70B, where Llama2-70B enjoys larger benefit of incorporating context through retrieval.

## 4.2 COMPARING TO OPENAI MODELS

To further understand how good is our best model, i.e., augmenting Llama2-70B-32k with retrieval, we also compare it to GPT-3.5-turbo(4k), GPT-3.5-turbo-16k and Davinci-003 on those seven datasets.[6] We found that Llama2-70B-32k-ret achieves better results than GPT-3.5-turbo-16k in terms of the average accuracy over seven datasets, while better than Davinci-003 (w/ 175B parameters) on the average over 4 tasks. This indicates Llama2-70B-32k with retrieval is a strong model for these long context tasks, and our conclusion is built on the state-of-the-art results.

We also report the retrieval augmented results for GPT3.5-turbo on MSQ, HQA and MFQA. For GPT3.5-turbo-4k, retrieval significantly improves the performance (avg from 37.08 to 41.15). For GPT3.5-turbo-16k, the average scores for retrieval (43.27) and non-retrieval (43.60) scores are close to each other which are both lower than our Llam2-70B-32k-ret results (44.51). Note that GPT3.5-turbo-16k is a blackbox API, we don't know how it is implemented, the model size as well as any preprocessing steps.

---

[6]For QMSum (QM), Qasper (QASP), NarrativeQA (NQA), QuALITY (QLTY), we used the test set from the ZeroSCROLLS leaderboard as the organizers have prepared the scores of GPT-3.5-turbo (4k) and Davinci-003 there.

| Model | Trec | SAMSum |
|---|---|---|
| GPT-3.5-turbo-16k | 68 | 41.7 |
| Llama2-70B | 73 | 46.5 |
| Llama2-70B-ret | 76 | 47.3 |

Table 6: Comparison of Llama2-70B to GPT-3.5-turbo-16k with two few-shot learning tasks from LongBench. Retrieval is helpful for few-shot learning as well.

### 4.3 Ablation on Different Retrievers

To investigate the impacts of different retrievers on top of Llama2-70B, we compare Dragon, Contriever, and OpenAI embeddings on top of Llama2-70B-4k and Llama2-70B-32k. The results in Table 4 confirms that our finding, i.e., *retrieval can boost the performance of both short context and long context LLMs*, is consistent across different retrievers.

### 4.4 Increasing the number of retrieved chunks

To study the impact of adding more retrieved chunks to the context, we increase the number of retrieved chunks from 5 to 20 using Dragon retriever and the results can be found in Table 5. We observe that for different sequence lengths, the best averaged results are obtained either from top 5 or top 10. Even if 20 chunks can still fit into the 16K and 32K context window (as shown in Figure 2), adding more chunks up to 20 is not helpful and will sometime hurt the performance. We believe this is related to the "lost in the middle" phenomenon (Liu et al., 2023) or the model is getting distracted by irrelevant information and therefore needs further research.

### 4.5 Retrieval for Few-shot Tasks

In addition to the zero-shot tasks of query-based summarization tasks and question answering tasks mentioned above, we further investigate the benefits of long context models for few-shot tasks using two additional datasets (Trec and SAMSum) from LongBench. We take the question from each dataset as the query and use it to search relevant QA pairs provided in the given few-shot examples. Table 6 shows that our best model Llama2-70B-32k-ret outperforms its non-retrieval Llama2-70B-32k baseline as well as GPT-3.5-turbo-16k by a large margin. It again confirms the benefits on using retrieval together with long context models.

## 5 Conclusion

In this work, we systematically study the retrieval-augmentation versus long context extension using the state-of-the-art LLMs after instruction tuning for various long context QA and query-based summarization tasks. After study, we have the following interesting findings: *i)* Retrieval largely boosts the performance of both 4K short context LLM and 16K/32K long context LLMs. *ii)* The 4K context LLMs with simple retrieval-augmentation can perform comparable to 16K long context LLMs, while being more efficient at inference. *iii)* After context window extension and retrieval-augmentation, the best model Llama2-70B-32k-ret can outperform GPT-3.5-turbo-16k and Davinci003 in terms of average score on a suit of downstream tasks with informative queries. Our study shed light on the promising direction of combining retrieval and long context techniques together to build better LLM.

## 6 Future Directions

There are many potential research directions that can be extended from this work. One direction is to develop advanced methods (e.g. memory or hierarchical attention) for existing pretrained large language models e.g. Llama2-70B, which is itself non-trivial. Also, further extending the context window to 64k and even longer would be a very interesting study for large 70B parameter models even though pre-training longer sequence requires much more computation. Lastly, how to mitigate the "lost-in-the-middle" phenomenon is an open research topic and continue pretraining with UL2 loss (Tay et al., 2022b) could be one potential solution.

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

# A APPENDIX

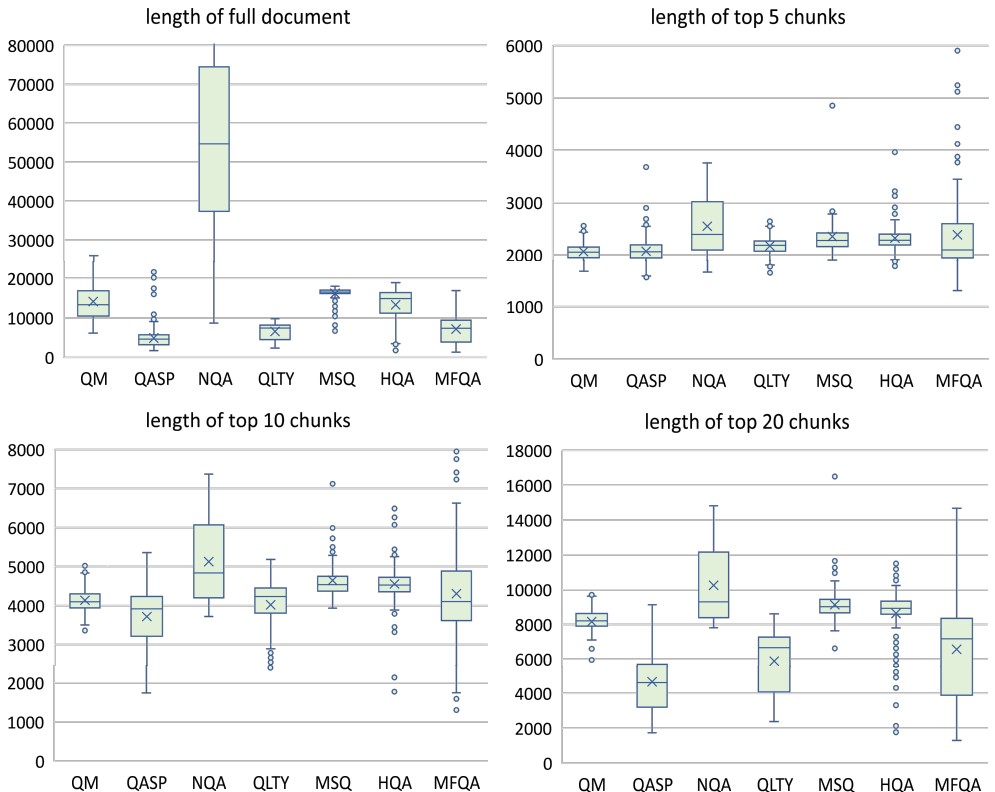

Figure 2: Token length distribution of the full document and the top-5, 10, 20 chunks of the seven datasets.

## A.1 EXAMPLE

We show an example below where the smaller model Llama2-7B fails to incorporate relevant context, while larger models with retrieval could successfully predict the correct answer.

| Chunk 1 | On September 18, 2015, the deluxe edition of the album was released containing live and instrumental tracks from the standard edition album, in addition to the single "Light" featuring Little Dragon. Critical reception ... Angelspit has toured with Angel Theory, Ayria, Ikon, KMFDM, Tankt and The Crüxshadows, and have also shared the stage with bands such as The Sisters of Mercy, Nitzer Ebb, Skinny Puppy and Front Line Assembly. They performed with Lords of Acid during a 22-date U.S. tour in March 2011 and **toured the United States with Blood on the Dance Floor in October 2011**. History Karl Learmont (ZooG) and Amelia Tan (Destroyx) met on an online zine forum. They shared an interest in zines and started the distro Vox Populis in 2002. |
|---|---|
| Chunk 2 | They then started making zines for themselves which became the lyrical inspiration for releases to follow. Angelspit was formed in 2003, and the duo then self-released their debut EP, Nurse Grenade on 3 October 2004. ... A video for the remix of "Sleep Now" was released on 2 October 2010. They released their third remix album, Carbon Beauty on 8 March 2011. This new remix album contains 3 new tracks as well as 10 remixes of tracks from the Hideous and Perfect album. A video for "Toxic Girl" was released on 13 April 2011, and a video for "Like It? |
| Chunk 3 | Passage 1: Blood on the Dance Floor (band) **Blood on the Dance Floor was an American electronic music group from Orlando, Florida, formed in 2006.** The group's longest standing lineup, from 2009 to 2016, consisted of Jesus David Torres also known as Dahvie Vanity (born 1984) and Jayy Von Monroe (born 1991). ... The CD was self-released in October 2008. Only 300 copies were made.Vanity and Ecstasy recorded the singles "Siq With a Q" and "Suicide Club" as a duo in 2008, and released three extended plays over the first half of 2009, I Scream I Scream, OMFG Sneak Peak, and Extended Play. |
| Chunk 4 | title: , source: Lick It!" was released on 27 July 2011. On 15 April 2011, Angelspit announced the addition of three new members: guitarist Valerie Gentile (Black Tape for a Blue Girl, The Crüxshadows), drummer Chris Kling (Hanzel und Gretyl, Mortiis) and videojammer The Liar. The new line-up of Angelspit released their fourth studio album, Hello My Name Is on 11 October 2011. Matt James replaced Chris Kling in early 2012, and former Crüxshadows guitarist George Bikos filled in for Valerie Gentile on the band's 2012 tour... |
| Chunk 5 | Vanity denied these allegations in a video.In April 2017, Vanity announced that there would be a new member, and that Blood on the Dance Floor would be returning on May 5 of that year without Jayy Von Monroe. ... On January 1, 2021, Vanity released a new version of Blood on the Dance Floor's "Bewitched" as "Bewitched Reimagined" featuring singer Sammy Beare. As of 2016, Jayy Von Monroe has continued to work as a drag monster under the name "The Dahli" and Vanity has continued to release music and merchandise by himself under both "The Most Vivid Nightmares" and "Dark Arts Official". |
| **Question:** | Angelspit toured the United States with an American electronic music duo from Orlando, Florida, that formed in what year? |
| **LLaMA2-7b-32k** | Angelspit toured the United States with Blood on the Dance Floor in October 2011. |
| **LLaMA2-7b-32k-ret** | Angelspit toured the United States with Blood on the Dance Floor in October 2011. |
| **LLaMA2-70b-32k** | 2011 |
| **LLaMA2-70b-32k-ret** | 2006 |
| **Ground truth** | 2006 |

Table 7: All models fail to get the correct answer of 2006, except LLaMA2-70b-32k-ret, which shows how retrieval together with long context can help derive the correct answer.

