# RETRIEVAL MEETS LONG CONTEXT LARGE LANGUAGE MODELS

## ABSTRACT

Extending the context window of large language models (LLMs) is getting popular recently, while the solution of augmenting LLMs with retrieval has existed for years. The natural questions are: *i) Retrieval-augmentation versus long context window, which one is better for downstream tasks? ii) Can both methods be combined to get the best of both worlds?* In this work, we answer these questions by studying both solutions using two state-of-the-art pretrained LLMs, i.e., a proprietary 43B GPT and LLaMA2-70B. Perhaps surprisingly, we find that shorter context window LLM with simple retrieval-augmentation at inference can perform close to longer context LLM finetuned via *positional interpolation* for question answering and query-based summarization tasks, while taking much less computation. More importantly, we demonstrate that retrieval can significantly improve the performance of LLMs regardless of their context window sizes. Our study provides general insights on the choice of retrieval-augmentation versus long context extension of LLM for practitioners.

## 1 INTRODUCTION

The long context large language models (LLM) have recently received a lot of attention in LLM production (e.g., Anthropic, 2023; OpenAI, 2023b), research community (e.g., Chen et al., 2023; Liu et al., 2023; Tworkowski et al., 2023), and open source community (e.g., Kaiokendev, 2023). Although the *approximate* attention methods have been studied for years (e.g., Tay et al., 2022) due to the quadratic time and memory complexities of self-attention mechanism in sequence length, the recent advance for long context LLMs with exact attention is mainly driven by the development of faster GPU with more memory and memory-efficient *exact* attention (Dao et al., 2022; Dao, 2023).

An alternative and long-standing solution for handling long context is *retrieval*. Specifically, the LLMs only read relevant context retrieved from a standalone retriever (e.g., Karpukhin et al., 2020; Wang et al., 2022; Lin et al., 2023), which runs orders of magnitudes faster than LLMs for selecting relevant context. Conceptually, the retrieval-augmented decoder-only LLM can be viewed as applying the sparse attention over its long context window, where the sparsity pattern is not predefined as Child et al. (2019) but determined by the standalone retriever. In other words, unretrieved context is treated as irrelevant and has zero-valued attention weights.

Given the surge of interest in long context LLM research and much more required computation at inference [1], it is still unclear for practitioners whether extending the context window of LLM provides higher accuracy than the retrieval augmentation for downstream tasks. Moreover, it would be compelling if we could combine the strength of both methods and achieve even higher accuracies. In this work, we attempt to answer the above questions through a comprehensive study.

Specifically, we make the following contributions:

1. We perform comprehensive study using two state-of-the-art LLMs, a proprietary 43B pretrained GPT and Llama 2-70B (Touvron et al., 2023b) on 7 downstream long context tasks, including single and multi document question answering (QA) as well as query-based summarization.

---

[1]For example, the price of GPT-4 with 32k context length is twice the 8k context model.

2. We demonstrate that retrieval-augmentation significantly improves the performance of 4K context LLMs. Perhaps surprisingly, we find this simple retrieval-augmented baseline can perform close to 16K long context LLMs, i.e., average score 29.32 vs. 29.45 by using GPT-43B, and 36.02 vs. 36.78 by using LLaMA2-70B, while using much less computation.

3. Furthermore, we demonstrate that the performance of long context LLM (i.e., 16K or 32K) can still be improved by retrieval, especially for the larger LLaMA2-70B. Our best long context model LLaMA2-70B-32k can be further enhanced by retrieval augmentation (avg. score improved from 40.9 to 43.6), and outperforms ChatGPT-3.5 (avg. score 41.1) by a margin.

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

- **Qasper (QASP)** (Dasigi et al., 2021) is a question answering dataset over NLP papers filtered from the Semantic Scholar Open Research Corpus (S2ORC) (Lo et al., 2020). Qasper contains abstractive, extractive, and yes/no questions, as well as unanswerable ones.

- **NarrativeQA (NQA)** (Kočiský et al., 2018) is an established question answering dataset over entire books from Project Gutenberg[2] and movie scripts from a list of websites. Summaries of the books and scripts obtained from Wikipedia were given to the annotators to produce question-answer pairs, resulting in approximately 30 questions and answers for each of the 1,567 books and scripts. Each question was answered by an additional annotator by providing two reference answers.

- **QuALITY (QLTY)** (Pang et al., 2022) is a multiple-choice question answering dataset over stories and articles sourced from several resources, such as Project Gutenberg and the Open American National Corpus[3]. 50% of the questions in QuALITY are labeled as *hard* to ensure the whole given document must be read slowly to conclude a correct answer, i.e., a skim of the document always yields wrong answers.

We take another three datasets from LongBench. (Bai et al., 2023).

---

[2]https://www.gutenberg.org/
[3]https://anc.org/

- **MuSiQue (MSQ)** (Trivedi et al., 2022) stands for Multihop Questions via Single-hop Question Composition aiming at multihop reasoning question answering. A bottom–up process of constructing multihop from single-hop questions allows systematic exploration of a large space of multihop candidates and greater control over which questions that are composed manually. In order to correctly generate the answers, LLMs require connected reasoning by reducing potential reasoning shortcuts, minimizing train-test leakage, and including harder distractor contexts. Thus, MuSiQue is significantly less cheatable via disconnected reasoning than previous datasets.

- **HotpotQA (HQA)** (Yang et al., 2018) is a Wikipedia-based question-answer dataset with several key features. First, multiple supporting documents are required to be read for answering and reasoning. Second, the questions are diverse and not constrained to any pre-existing knowledge bases. Third, sentence-level supporting are provided with strong supervision to support LLM's requirement for reasoning. Finally, new types of factoid comparison questions are provided to test LLMs' ability to extract and compare various entity properties in text.

- **MultiFieldQA-en (MFQA)** (Bai et al., 2023) was manually curated to better test the model's long context understanding ability across diverse fields. Documents and articles from multiple sources, including legal documents, government reports, encyclopedias, and academic papers are collected. Ph.D. students were asked to annotate the questions and answers for each article. The evidences are fairly randomly located in the documents to avoid biases that might occur at the beginning or ending of the documents.

The full details of the dataset can be found in Table 1. We can see that our evaluation datasets have a wide range of average document length from 4.9k (QASP) to 84k (NQA). Therefore, for the baseline model without retrieval, we truncate the document accordingly to fit into the input sequence length.

Following the official metrics, we report the geometric mean of ROUGE scores (i.e., ROUGE-1/2/L) (Lin, 2004) for QM, the exact matching (EM) score for QLTY, and F1 scores for the remaining five datasets QASP, NQA, MSQ, HQA and MFQA.

### 3.3 CONTEXT WINDOW EXTENSION

We extend the context window length with position interpolation method (Chen et al., 2023), as it is simple and effective for RoPE embeddings. We extend the 4K context window to 16K for GPT-43B. For LLaMA2-70B, we extend its 4K context window to 16K and 32K. We follow Chen et al. (2023) and finetune both LLMs on Pile dataset (Gao et al., 2021) with batch size as 128, constant learning rate of 5e-6 to adapt the position embeddings.

### 3.4 RETRIEVAL

For the retriever, we experimented with three retrievers: (1) Dragon (Lin et al., 2023) as it achieves state-of-the-art results on both supervised and zero-shot information retrieval benchmarks (Thakur et al., 2021). Dragon is a dual encoder model that consists of a query encoder and a context encoder. (2) a widely used Contriever model (Izacard et al., 2021). Following the MoCo technique (He et al., 2020), Contriever used a simple contrastive learning framework to pre-train models for information retrieval. It was trained without supervision and achieved competitive results with BM25 for R@100 on the BEIR benchmark (Thakur et al., 2021), and (3) OpenAI embedding[4]. For the OpenAI embedding model, we use the latest "text-embedding-ada-002" as recommended by OpenAI. It accepts 8,191 maximum input tokens for one sequence with an output vector of 1,536 dimensions. The cosine similarities are then computed between the questions and the list of contexts for retrieval ranking.

To use these retrievers, we first chunk each context document with 300 words, and then we encode both the questions and all chunks independently with corresponding encoder. The most relevant N chunks, ranked by the dot product of the question embedding and chunk embedding, are then combined as the context of the prompt for generation. Table 1 shows the statistics of the top N retrieved chunks while Figure 1 gives more details of the token length distribution of all seven datasets.

---

[4]https://platform.openai.com/docs/guides/embeddings

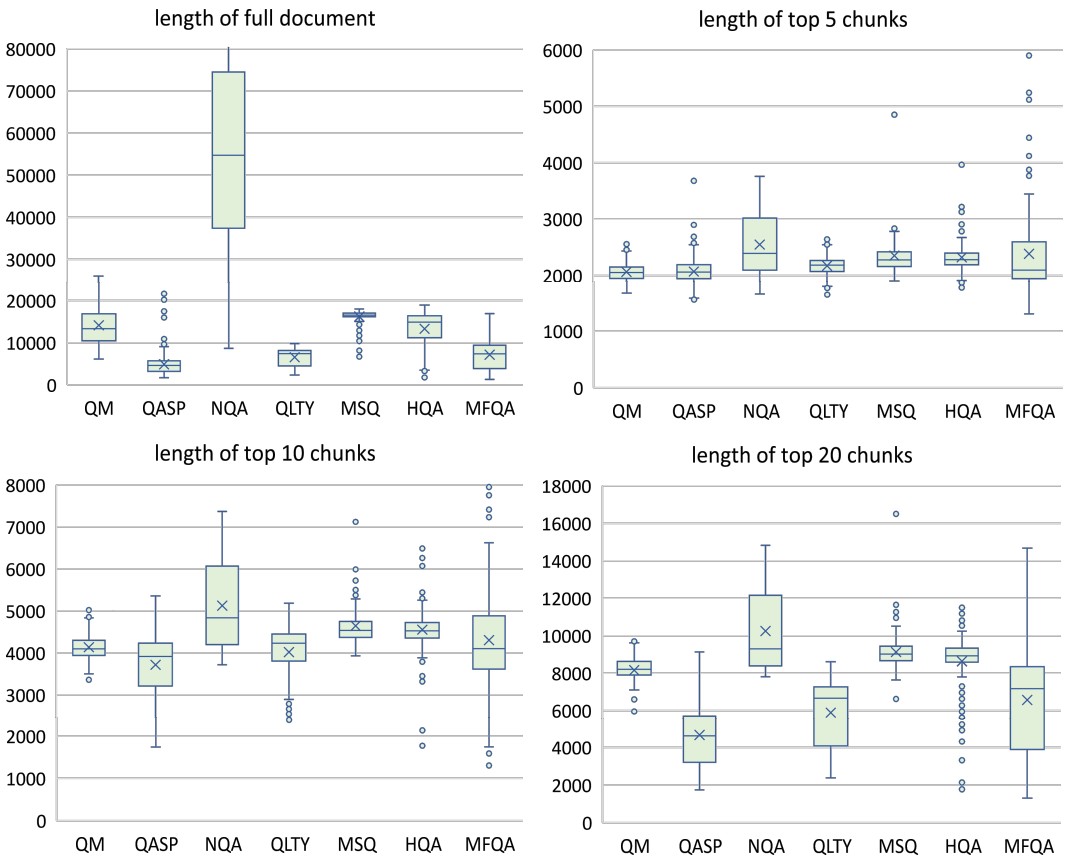

Figure 1: Token length distribution of the full document and the top-5, 10, 20 chunks of the seven datasets.

Note that, some dataset like Qasper (QASP) is relatively short and don't have up to 20 chunks, so the average length of top-10 chunks and top-20 chunks are close. We can see that top-5 chunks can all fit into 4k sequence length (except few outliers) while top-10 and top-20 chunks can fit into 16k sequence length.

### 3.5 INSTRUCTION TUNING

To train the pretrained LLMs to follow instructions for question answering or text summarization, we also performed instruction tuning. We first construct a blend of instruction tuning datasets consisting of 102K training samples from the Soda dataset (Kim et al., 2022), ELI5 dataset (Fan et al., 2019), FLAN dataset (Wei et al., 2021) , Open Assistatant dataset (Köpf et al., 2023), to adapt both GPT-43B and LLaMA2-70B to follow instructions. In terms of the template, we use "System: {System}\n\nUser: {Question}\n\nAssistant: {Answer}" as the format to support multi-turn dialogue training. As all of the tasks contain the context information for reasoning over at inference time, we add the context before the dialogue, i.e. "System: {System}\n\n{Context}\n\nUser: {Question}\n\nAssistant: {Answer}".

We finetune the LLM by taking the loss only on the answer part with batch size 128 and learning rate of 5e-6 for 1000 steps. For the rest of the paper, results are all reported using the instruction tuned chat model on top of the foundation GPT-43B and LLaMA2-70B.

## 4 RESULTS

In this section, we report the results and provide detailed analysis.

| Model | Seq len. | Avg. | QM | QASP | NQA | QLTY | MSQ | HQA | MFQA |
|---|---|---|---|---|---|---|---|---|---|
| GPT-43B | 4k | 26.44 | 15.56 | 23.66 | 15.64 | 49.35 | 11.08 | 28.91 | 40.90 |
| + ret | 4k | 29.32 | 16.60 | 23.45 | 19.81 | 51.55 | 14.95 | 34.26 | 44.63 |
| GPT-43B | 16k | 29.45 | 16.09 | 25.75 | 16.94 | 50.05 | 14.74 | 37.48 | 45.08 |
| + ret | 16k | **29.65** | 15.69 | 23.82 | 21.11 | 47.90 | 15.52 | 36.14 | 47.39 |
| LLaMA2-70B | 4k | 31.61 | 16.34 | 27.70 | 19.07 | 63.55 | 15.40 | 34.64 | 44.55 |
| + ret | 4k | 36.02 | 17.41 | 28.74 | 23.41 | 70.15 | 21.39 | 42.06 | 48.96 |
| LLaMA2-70B | 16k | 36.78 | 16.72 | 30.92 | 22.32 | **76.10** | 18.78 | 43.97 | 48.63 |
| + ret | 16k | 37.23 | **18.70** | 29.54 | 23.12 | 70.90 | 23.28 | 44.81 | 50.24 |
| LLaMA2-70B | 32k | 37.36 | 15.37 | **31.88** | 23.59 | 73.80 | 19.07 | 49.49 | 48.35 |
| + ret | 32k | **39.60** | 18.34 | 31.27 | **24.53** | 69.55 | **26.72** | **53.89** | **52.91** |

Table 2: Comparison of model variants (GPT-43B, LLaMA2-70B) with sequence length ranging from 4k to 32k under seven datasets. "ret" denotes using the best retriever (Dragon or Contriever) and here we used top-5 for the retriever.

## 4.1 MAIN RESULTS

In Table 2, we compare different model variants with context lengths ranging from 4K to as long as 32K using GPT-43B and LLaMA2-70B. First, we find that baseline models without retrieval for 4k sequence length achieve the worst results for both GPT-43B and LLaMA2-70B. This is because the minimum average sequence length of all seven tasks is more than 4096, which exceeds the context window of the foundation models and therefore gets truncated without considering the semantics of the context. As a result, retrieval is especially helpful for 4K context LLMs e.g., LLaMA2-70B-4K is improved from 31.61 to 35.73 while GPT-43B-4K is improved from 26.44 to 29.32. Second, we observe that HotpotQA (HQA) especially favors long sequence models as the score improves from 34.64 to 43.97 for LLaMA2-70B and from 28.91 to 37.48 for GPT-43B when the sequence length increases from 4k to 16k. This is because Hotpot QA is a multi-hop dataset where the questions are not hard to answer but all intermediate hops are necessary to get correct answer. Therefore, long context are beneficial to increase the probability of incorporating all intermediate hops.

It is quite interesting that the retrieval-augmented long context LLM (e.g., 16K and 32K) can obtain better results than retrieval-augmented 4K context LLM, even they are feed with the same top 5 chunks of evidence. We hypothesize this interesting observation is related to the "lost-in-the-middle" phenomenon (Liu et al., 2023), where the LLMs has such "U-shaped" performance curve. Specifically, LLMs are better at utilizing relevant information that occurs at the beginning or end of its input context window. Due to this reason, the 4K context LLM tends to ignore the information in the middle of 4K input, while 32K context LLM tend to ignore the information in the middle of 32K input. From Figure 1, the length of top 5 chunks is about 2K tokens, which can be in the middle and ignored by 4K context LLM, but is only at the beginning part of 16K and 32K context and may not be ignored by the 16K or 32K context LLM.

Note that, we have very different observation from the conclusion drawn from LongBench work (Bai et al., 2023): *"Retrieval brings improvement for model with weak ability on long contexts, but the performance still lags behind models that have strong long context understanding capability"*. Here, we demonstrate retrieval can significantly improve the performance of both GPT-43B and LLaMA2-70B regardless their context window size. For example, our best retrieval-augmented LLaMA2-70B-32k-ret outperforms its baseline w/o retrieval by a margin, i.e., 39.60 vs. 37.36. We think the major reason for such different conclusion is that Bai et al. (2023) uses much smaller LLM with 6B and 7B parameters, which usually has relatively worse zero-shot capability to incorporate the retrieved chunked context. In contrast, the larger instruction tuned LLMs like LLaMA2-70B has much stronger zero-shot capability to incorporate retrieved evidence. This observation is becoming more clear when one compares the gain of retrieval-augmentation between GPT-43B and LLaMA2-70B, where LLaMA2-70B enjoys larger benefit of incorporating context through retrieval.

| Model | Avg. | QM* | QASP* | NQA* | QLTY* | MSQ | HQA | MFQA |
|---|---|---|---|---|---|---|---|---|
| Davinci003 (175B) | - | 16.9* | 52.7* | 24.6* | 69.0* | - | - | - |
| ChatGPT-3.5 | 41.1 | 15.6* | 49.3* | 25.1* | 66.6* | 26.9 | 51.6 | 52.3 |
| LLaMA2-70B-32k | 40.9 | 15.6 | 45.9 | 28.4 | 79.6 | 19.1 | 49.5 | 48.4 |
| LLaMA2-70B-32k-ret | **43.6** | 18.5 | 46.3 | 31.5 | 75.6 | 26.7 | 53.9 | 52.9 |

Table 3: Comparison of our best retrieval-augmented LLaMA2-70B with ChatGPT-3.5 and Davinci-003 (w/ 175B parameters). * denotes the number (for ChatGPT-3.5 and Davinci-003) or task is taken from Zero Scroll leaderboard. For QMSum (QM), Qasper (QASP), NarrativeQA (NQA), QuALITY (QLTY), we used the test set from the Zero Scroll leaderboard as the organizers have prepared the scores of ChatGPT and Davinci-003 there.

| Seq len | Setting | Avg. | QM | QASP | NQA | QLTY | MSQ | HQA | MFQA |
|---|---|---|---|---|---|---|---|---|---|
| 4k | baseline (w/o ret) | 31.61 | 16.34 | 27.70 | 19.07 | 63.55 | 15.40 | 34.64 | 44.55 |
| | Dragon | 35.73 | 18.14 | 29.20 | 23.39 | 70.30 | 20.09 | 41.54 | 47.45 |
| | Contriever | **36.02** | 17.41 | 28.74 | 23.41 | 70.15 | 21.39 | 42.06 | 48.96 |
| | OpenAI-embedding | 35.79 | 17.76 | 28.85 | 23.57 | 70.70 | 19.92 | 41.76 | 47.99 |
| 32k | baseline (w/o ret) | 37.36 | 15.37 | 31.88 | 23.59 | 73.80 | 19.07 | 49.49 | 48.35 |
| | Dragon | **39.60** | 18.34 | 31.27 | 24.53 | 69.55 | 26.72 | 53.89 | 52.91 |
| | Contriever | 38.85 | 17.60 | 31.56 | 23.88 | 69.00 | 26.61 | 49.65 | 53.66 |
| | OpenAI-embedding | 39.34 | 18.24 | 32.07 | 24.36 | 69.45 | 24.90 | 51.64 | 54.75 |

Table 4: Comparisons of adding top 5 retrieved chunks from different retrievers to the context under LLaMA2-70B. Public available retriever can be better than OpenAI-embedding.

| Seq len | Setting | Avg. | QM | QASP | NQA | QLTY | MSQ | HQA | MFQA |
|---|---|---|---|---|---|---|---|---|---|
| 4k | base | 31.61 | 16.34 | 27.70 | 19.07 | 63.55 | 15.40 | 34.64 | 44.55 |
| | top-5 | **35.73** | 18.14 | 29.20 | 23.39 | 70.30 | 20.09 | 41.54 | 47.45 |
| | top-10 | 34.62 | 16.54 | 28.67 | 24.38 | 68.70 | 19.00 | 42.18 | 42.84 |
| | top-20 | 34.61 | 16.52 | 28.67 | 24.38 | 68.70 | 19.00 | 42.18 | 42.84 |
| 16k | base | 36.78 | 16.72 | 30.92 | 22.32 | 76.10 | 18.78 | 43.97 | 48.63 |
| | top-5 | 37.23 | 18.70 | 29.54 | 23.12 | 70.90 | 23.28 | 44.81 | 50.24 |
| | top-10 | **38.31** | 18.41 | 30.20 | 25.53 | 73.60 | 22.78 | 47.72 | 49.91 |
| | top-20 | 36.61 | 17.26 | 29.60 | 25.81 | 72.30 | 22.69 | 41.36 | 47.23 |
| 32k | base | 37.36 | 15.37 | 31.88 | 23.59 | 73.80 | 19.07 | 49.49 | 48.35 |
| | top-5 | **39.60** | 18.34 | 31.27 | 24.53 | 69.55 | 26.72 | 53.89 | 52.91 |
| | top-10 | 38.98 | 17.71 | 30.34 | 25.94 | 70.45 | 22.80 | 55.73 | 49.88 |
| | top-20 | 38.38 | 16.36 | 30.42 | 24.42 | 69.60 | 24.51 | 54.67 | 48.65 |

Table 5: Comparisons of adding top-5/10/20 retrieved chunks to the context under 4k, 16k, and 32k input sequence lengths using LLaMA2-70B. More context does not always give better results.

## 4.2 COMPARING TO OPENAI MODELS

To further understand how good is our best model using LLaMA2-70B-32k with retrieval, we also compare it to ChatGPT-3.5 and Davinci-003 on those seven datasets. [5] We found that LLaMA2-70B-32k-ret achieves better results than ChatGPT-3.5 in terms of the average accuracy over seven datasets, while better than Davinci-003 (w/ 175B parameters) on the averge over 4 tasks. This indicates LLaMA2-70B-32k with retrieval is a strong model for these long context tasks, and our conclusion is built on the state-of-the-art results.

---

[5] For QMSum (QM), Qasper (QASP), NarrativeQA (NQA), QuALITY (QLTY), we used the test set from the Zero Scroll leaderboard as the organizers have prepared the scores of ChatGPT and Davinci-003 there.

### 4.3 ABLATION ON DIFFERENT RETRIEVERS

To investigate the impacts of different retrievers on top of LLaMA2-70B, we compare Dragon, Contriever, and OpenAI embeddings on top of LLaMA2-70B-4k and LLaMA2-70B-32k. The results in Table 4 confirms that our finding, i.e., *retrieval can boost the performance of both short context and long context LLMs*, is consistent across different retrievers. Also, we observe that public available retrievers can do better than the commercially closed OpenAI embeddings.

### 4.4 INCREASING THE NUMBER OF RETRIEVED CHUNKS

To study the impact of adding more retrieved chunks to the context, we increase the number of retrieved chunks from 5 to 20 using Dragon retriever and the results can be found in Table 5. We observe that for different sequence lengths, the best averaged results are obtained either from top 5 or top 10. Even if 20 chunks can still fit into the 16K and 32K context window (as shown in Figure 1), adding more chunks up to 20 is not helpful and will sometime hurt the performance. We believe this is related to the "lost-in-the-middle" phenomenon (Liu et al., 2023) or the model is getting distracted by irrelevant information and therefore needs further research.

## 5 CONCLUSION

In this work, we systematically study the retrieval-augmentation versus long context extension using the state-of-the-art LLMs after instruction tuning for various long context QA and query-based summarization tasks. After study, we have the following interesting findings: *i)* Retrieval largely boosts the performance of both 4K short context LLM and 16K/32K long context LLMs. *ii)* The 4K context LLMs with simple retrieval augmentation can perform close to 16K long context LLMs, while being more efficient at inference. *iii)* After context window extension and retrieval-augmentation, the best model LLaMA2-70B-32k-ret can outperform ChatGPT-3.5 and Davinci003 by a margin. Our study shed light on the promising direction of combining retrieval and long context techniques together to build better LLM.