# OpenReview forum: "Retrieval meets Long Context Large Language Models"
_ICLR.cc/2024/Conference — ICLR 2024 poster_

### Official Review · Reviewer_L1CH · 2023-10-31

**Soundness:** 3 good
**Presentation:** 4 excellent
**Contribution:** 3 good
**Rating:** 8
**Confidence:** 3

**Summary:**

The concept of augmenting language models with retrieval has been widely explored for several years. Recently, there has been a growing interest in expanding the context window of transformers and large language models (LLMs). This research paper focuses on two key research questions related to retrieval augmentation and context window extension:

1. Which approach is more effective for downstream tasks - retrieval augmentation or a longer context window?
2. Is it possible to combine these methods to achieve better results?

To address these questions, the authors of this paper examine two pretrained LLMs - a 43B GPT model trained by themselves and LLaMA2-70B. Through extensive experimentation, they find that a shorter context window LLM with simple retrieval augmentation during inference can yield comparable results to a longer context LLM that is fine-tuned using positional interpolation in two specific downstream tasks, namely question answering and query-based summarization. Additionally, the authors note that retrieval can significantly enhance the performance of LLMs, regardless of the size of their context window. The findings of this study offer valuable insights to practitioners, helping them make informed decisions regarding the choice between retrieval augmentation and extending the context window of LLMs.

**Strengths:**

1. This paper has a clear motivation and investigates a significant topic on large language models (retrieval augmentation versus context window extension). Existing research effort on this topic is limited, except [1]. Instead of using black box API in [1] (e.g. GPT-3.5-Turbo), this paper trained their own 43B GPT model for comparisons.
2. This paper is well-written and the presentation is clear;
3. Extensive experiments have been conducted on 7 downstream long context tasks, including single and multi-document question answering and query-based summarization. Two different sizes of LLMs and Three different retrievers are examined on eight tasks.


Reference:

[1] Longbench: A bilingual, multitask benchmark for long context understanding. Arxiv 2023

**Weaknesses:**

1. The authors of this paper mainly focus on larger LLMs, specifically those with more than 40B parameters. They include two specific LLMs in their study: the 43B GPT and the 70B LLaMA. It is worth noting that most existing efforts in this field only report results on smaller models, such as the 7B model. It would be beneficial if the authors could further investigate the relationship between the size of the LLMs and the tradeoff between retrieval and the context window. The authors propose hypotheses to explain why concurrent work has produced different findings, particularly regarding the usefulness of retrieval for the Llama2-7B-chat-4k model with a 4K context window. It would be better if the authors could provide empirical evidence to support these hypotheses.

2. In terms of context window extension, this paper only utilizes the method mentioned in reference [2] for both LLMs. It would be interesting to explore whether the findings would be influenced by different methods of extending the context window.

3. This paper focuses primarily on question answering and summarization tasks, in contrast to Longbench[1], which encompasses a variety of tasks, including few-shot learning, synthetic tasks, and code completion.

4. The paper lacks a discussion on potential limitations and future directions. For instance, the authors do not address the potential limitations of combining retrieval augmentation and context window extension, such as the possibility of exacerbating the "lost-in-the-middle" phenomenon [3]. While one of the key findings of this study is that combining retrieval and long-context techniques improves LLM performance, the authors could suggest avenues for future research on how to better integrate these techniques together.

Reference:

[2] Extending context window of large language models via positional interpolation. Arxiv 2023
[3] Lost in the middle: How language models use long contexts. Arxiv 2023

**Questions:**

See above.

---

> ### Author Response · Authors · 2023-11-23
> **Rebuttal by Authors**
>
> Many thanks for your detailed review. We will address your comments in the following
>
> > 1. The authors of this paper mainly focus on larger LLMs, specifically those with more than 40B parameters. They include two specific LLMs in their study: the 43B GPT and the 70B LLaMA. It is worth noting that most existing efforts in this field only report results on smaller models, such as the 7B model. It would be beneficial if the authors could further investigate the relationship between the size of the LLMs and the tradeoff between retrieval and the context window. The authors propose hypotheses to explain why concurrent work has produced different findings, particularly regarding the usefulness of retrieval for the Llama2-7B-chat-4k model with a 4K context window. It would be better if the authors could provide empirical evidence to support these hypotheses.
>   - Following your suggestion, we have conducted the experiments for llama2-7b and the results are presented below.
>
> |        | Seq len             | Avg.          | QM | QASP | NQA | QLTY | MSQ | HQA | MFQA|
> | ------ | --------------- | ------- | -- | - | - | - | - | - | - |
> | Llama2-7B            | 4k   | 22.65  | 14.25  |	22.07 |	14.38  | 40.90 | 8.66  |	23.13  | 35.20|
> | Llama2-7B (+ret)  | 4k    | 26.04 | 16.45  |	22.97 |	18.18  | 43.25 | 14.68 | 26.62 |40.10 |
> | Llama2-7B            | 32k  | 28.20 | 16.09 | 23.66   |19.07   | 44.50 | 15.74 | 31.63 |46.71 |
> | Llama2-7B (+ret)  | 32k  | 27.63 | 17.11  | 	23.25  |19.12  | 43.70 | 15.67 | 29.55 | 45.03 |
>
> For 4k sequence length, we observed retrieval can still have significant improvements while for 32k sequence length, adding retrieval is getting worse, which aligns with the findings in LongBench (Bai et al. (2023) for ChatGLM2-6B-32k. We believe the reason is that for Llama2-7B-chat-32 and ChatGLM2-6B-32k, their retrieval-augmented models have limited zero-shot capability of incorporating retrieved chunks of context, due to the smaller size. As a result, retrieval is not helpful for both Llama2-7B-32k and ChatGLM2-6B-32k, which is different from large LLMs like Llama2-70B-32k.  We add these discussion in the updated draft.
>
> > 2. In terms of context window extension, this paper only utilizes the method mentioned in reference [2] for both LLMs. It would be interesting to explore whether the findings would be influenced by different methods of extending the context window.
>   - We agree with your comment in general. However, making advanced methods (e.g. memory or hierarchical attention) works for existing pretrained large language models e.g. Llama2-70B, is itself non-trivial. Further Pre-training the model with longer sequence requires much more computation. We add those into our future work section.
>
> > 3. This paper focuses primarily on question answering and summarization tasks, in contrast to Longbench[1], which encompasses a variety of tasks, including few-shot learning, synthetic tasks, and code completion.
>   - Thanks for your suggestion. We conduct few-shot experiment on Trec and SAMsum and the results are as follow:
>
> |                                | Trec   | SAMSum |
> |:-------------:               |:-------:|:-------------:|
> | GPT3.5-turbo-16k   |    68  |    41.7        |
> | Llama2-70b-32k     |    73   |    46.48      |
> | Llama2-70b-32k-ret |  76    |    47.31      |
>
> Our best model Llama2-70B-32k-ret outperforms its non-retrieval Llama2-70B-32k baseline as well as GPT-3.5-turbo-16k by a large margin. It again confirms the benefits of using retrieval together with long context models. We have included this part in Section 4.5
>
> > 4. The paper lacks a discussion on potential limitations and future directions. For instance, the authors do not address the potential limitations of combining retrieval augmentation and context window extension, such as the possibility of exacerbating the "lost-in-the-middle" phenomenon [3]. While one of the key findings of this study is that combining retrieval and long-context techniques improves LLM performance, the authors could suggest avenues for future research on how to better integrate these techniques together.
>   - Thanks for your suggestion, one possible solution to mitigate the “lost-in-the-middle” phenomenon is to further train the model with UL2 loss. In terms of future research, we add a future work section so that others can build on top of our findings.

---

### Official Review · Reviewer_FNKh · 2023-11-01

**Soundness:** 3 good
**Presentation:** 3 good
**Contribution:** 2 fair
**Rating:** 6
**Confidence:** 4

**Summary:**

In this paper, the authors systematically study the effectiveness of utilizing retrieval-augmentation and long-context tension in LLMs for downstream tasks. They conduct experiments using two large foundational models and evaluate their performance on seven downstream tasks that require longer context understanding. Compared to a previous study (Bai et al. 2023),  this work employs larger LLMs, leading to a different conclusion: retrieval significantly enhances the performance of LLMs, regardless of the size of their context window. Furthermore, the authors observe that larger models, such as LLaMA2-70B, tend to benefit more from retrieved evidence.

**Strengths:**

- The findings from these extensive empirical experiments are intriguing and have the potential to serve as inspiration for future research in this domain.
- The authors demonstrate that retrieval and extended-context approaches can significantly benefit LLMs, especially when they are of extremely large size.
- The study incorporates further analyses that clarify the impact of factors such as the number of chunks, context length, and retrievers on the performance of downstream tasks.

**Weaknesses:**

- While the idea is straightforward, the novelty is limited.
- The 70B model consistently exhibits improvement when utilizing retrieval or longer context. In contrast, the 43B model shows inconsistent improvement. Consequently, drawing a definitive conclusion regarding when to employ retrieval in relation to specific downstream tasks and model sizes becomes challenging.
- Given that the conclusion differs from prior work that utilizes smaller models, establishing an apple-to-apple comparison demonstrating instances where the same example could fail in the smaller model but succeed in the larger model would enhance the persuasiveness of the findings.

**Questions:**

- Do you observe different conclusions when using a smaller LLM (i.e., Llama2-7B and ChatGLM2-6B) as in Bai et al. 2023?

---

> ### Author Response · Authors · 2023-11-23
> **Rebuttal by Authors**
>
> Many thanks for your detailed review. We will address your comments in the following.
>
> Weaknesses:
> > 1. “While the idea is straightforward, the novelty is limited.”
>   - The focus of this paper is not proposing any novel idea for LLMs research. Intead, it investigates and combines two important trends in LLMs i.e., retrieval-augmented generation (RAG) and long context LLMs. The obtained results are significant; for example, our best model, retrieval-augmented LLaMA2-70B with 32K context window, outperforms GPT-3.5-turbo-16k and Davinci003 in terms of average score on seven long context tasks including question answering and query-based summarization (see the updated Table 3 in the updated manuscript). The conclusion and insights are very useful for the practitioners; for example, RAG and long context LLMs should be combined to obtain the best results. As a result, we believe this work is an important, timely and very useful contribution to the field.
>   - As your comment on the Strengths section pointed out, “the findings from these extensive empirical experiments are intriguing and have the potential to serve as inspiration for future research in this domain”. We believe the important findings are as important as novel ideas.
>
> > 2. “The 70B model consistently exhibits improvement when utilizing retrieval or longer context. In contrast, the 43B model shows inconsistent improvement. Consequently, drawing a definitive conclusion regarding when to employ retrieval in relation to specific downstream tasks and model sizes becomes challenging.”
>   - Note that both 70B and 43B models have consistent improvement in terms of **average** score across 7 downstream tasks when utilizing retrieval or long context extension, as shown in Table 2.  For 43B LLM, i) We observe improvements from baseline 43B-4k to retrieval-augmented 43B-4k on 6 out of 7 downstream tasks, except on QASP with small difference (baseline 23.66 vs. retrieval-augmented 23.45). ii) We also observe consistent improvement from baseline 43B-4k to long context 43B-16k on all 7 downstream tasks. iii ) The retrieval-augmented 43B-16k still outperforms its baseline 43B-16k in terms of average score (29.65 vs. 29.45), but we see variations on individual downstream tasks here partially due to the inherent variance exists in zero-shot evaluation. It is worth mentioning that we aim to draw conclusions for the average setting instead of specific downstream task across the paper.
>
> > 3. “Given that the conclusion differs from prior work that utilizes smaller models, establishing an apple-to-apple comparison demonstrating instances where the same example could fail in the smaller model but succeed in the larger model would enhance the persuasiveness of the findings.” and “Do you observe different conclusions when using a smaller LLM (i.e., Llama2-7B and ChatGLM2-6B) as in Bai et al. 2023?”
>   - Bai et al. (2023) find that retrieval is only helpful for Llama2-7B-chat-4k with 4K context window, but not helpful for long context model ChatGLM2-6B-32k.  We discussed this in Related Work Section 2.4 of submission. To highlight it, we move it to the beginning of related work Section 2.1.
>   - Per your suggestion, we have added the results using Llama2-7B in the following Table and updated draft. One can actually draw similar conclusions to Bai et al. (2023) for small 7b models. We think the underlying reasons are: i) For Llama2-7B-chat-4k, its short context length is the bottleneck for long context tasks. Thus, retrieval-augmentation largely improves the results. ii) For Llama2-7B-chat-32 and ChatGLM2-6B-32k, the context length bottleneck has been mostly removed. However, their retrieval-augmented models have limited zero-shot capability of incorporating retrieved chunks of context, due to the smaller size. As a result, retrieval is not helpful for both Llama2-7B-32k and ChatGLM2-6B-32k, which is different from large LLMs like Llama2-70B-32k. We also add this analysis and results into our updated manuscript.
>
>
> |        | Seq len             | Avg.          | QM | QASP | NQA | QLTY | MSQ | HQA | MFQA|
> | ------ | --------------- | ------- | -- | - | - | - | - | - | - |
> | Llama2-7B            | 4k   | 22.65  | 14.25  |	22.07 |	14.38  | 40.90 | 8.66  |	23.13  | 35.20|
> | Llama2-7B (+ret)  | 4k    | 26.04 | 16.45  |	22.97 |	18.18  | 43.25 | 14.68 | 26.62 |40.10 |
> | Llama2-7B            | 32k  | 28.20 | 16.09 | 23.66   |19.07   | 44.50 | 15.74 | 31.63 |46.71 |
> | Llama2-7B (+ret)  | 32k  | 27.63 | 17.11  | 	23.25  |19.12  | 43.70 | 15.67 | 29.55 | 45.03 |
>
>   - We also add concrete examples in Appendix where the smaller model fails to incorporate relevant context, while larger models could successfully do that.

---

> > ### Comment · Reviewer_FNKh · 2023-11-23
> >
> > Thanks for the authors' response! I have changed my score accordingly.  Good luck !

---

### Official Review · Reviewer_xWjb · 2023-11-01

**Soundness:** 3 good
**Presentation:** 3 good
**Contribution:** 2 fair
**Rating:** 8
**Confidence:** 4

**Summary:**

The paper presents a detailed comparison of different methods of handling long contexts by 1) using retrieval augmentation 2) increasing the context length of LLMs using positional interpolation. The papers conducts experiments on 7 datasets and show that retrieval can significantly improve the performance of LLMs. The paper also shows results on increasing the number of retrieved context and compares different retrievers.

**Strengths:**

The paper performs comprehensive experiments on two LLMs with up to 32K context, demonstrating the benefits of retrieval even for long context models. The paper gives good insights about retrieval augmentation confirming a simple but effective alternative to expensive context scaling of huge models. The paper is well-written and easy to understand.

**Weaknesses:**

The paper has limited novelty. The majority of the paper compares to longer context LLM finetuned via positional interpolation, it may be better to have some results with the other context expansion techniques and with models that are natively trained with larger context lengths to possibly remove that source of error.

**Questions:**

If the chunking is 300 words (tokens ?), why is there so much variance in the top-5, top-10, shouldn't the top-5 context length consistently be around 1500 tokens? If the variability is due to chunking based on words and not tokens, why is that the case ? and why was 300 specifically chosen?

---

> ### Author Response · Authors · 2023-11-23
> **Rebuttal by Authors**
>
> Many thanks for your detailed review. We will address your comments in the following.
>
> Weaknesses:
> > 1. The paper has limited novelty.
>   - The focus of this paper is not proposing any novel idea for LLMs research. Intead, it investigates and combines two important trends in LLMs i.e., retrieval-augmented generation (RAG) and long context LLMs. The obtained results are significant; for example, our best model, retrieval-augmented LLaMA2-70B with 32K context window, outperforms GPT-3.5-turbo-16k and Davinci003 in terms of average score on seven long context tasks including question answering and query-based summarization (see the updated Table 3 in the updated manuscript). The conclusion and insights are very useful for the practitioners; for example, RAG and long context LLMs should be combined to obtain the best results. As a result, we believe this work is an important, timely and very useful contribution to the field.
>   - As your comment on the Strengths section pointed out, “The paper gives good insights about retrieval augmentation confirming a simple but effective alternative to expensive context scaling of huge models.”  We believe the important findings are as important as novel ideas.
>
> > 2. The majority of the paper compares to longer context LLM finetuned via positional interpolation, it may be better to have some results with the other context expansion techniques and with models that are natively trained with larger context lengths to possibly remove that source of error.
>   - We agree with your comment in general. However, making advanced methods (e.g. memory or hierarchical attention) works for existing pretrained large language models e.g. Llama2-70B, is itself non-trivial. Further pre-training the model with longer sequence length requires much more computation. We leave those as future work.
>
> > 3. If the chunking is 300 words (tokens ?), why is there so much variance in the top-5, top-10, shouldn't the top-5 context length consistently be around 1500 tokens? If the variability is due to chunking based on words and not tokens, why is that the case ? and why was 300 specifically chosen?
>   - Following existing work [1,2], we chunk the document by words. And yes, the variability is due to chunking based on words. The reason is that many of those texts contain symbols, academic terms, numbers, as well as noises, which causes lots of variance in the tokenization. As our retriever accepts up to 512 tokens, we choose 300 as a high enough number to fit into the length limit as well as accommodating more continuous text for the context.
>
> [1] Bai, Yushi et al. “LongBench: A Bilingual, Multitask Benchmark for Long Context Understanding.” ArXiv abs/2308.14508 (2023): n. pag.
>
> [2] Karpukhin, Vladimir, et al. "Dense passage retrieval for open-domain question answering." arXiv preprint arXiv:2004.04906 (2020).

---

### Official Review · Reviewer_MoDw · 2023-11-03

**Soundness:** 3 good
**Presentation:** 3 good
**Contribution:** 3 good
**Rating:** 6
**Confidence:** 3

**Summary:**

This paper studies tasks that require reading / reasoning over long documents, and considers two commonly used approaches: retrieval augmentation and long-context LLMs. By experimenting on 7 long-context tasks from the Scroll and LongBench benchmarks, this paper first compares the two solutions and finds that a retrieval-augmented LLM with standard context windows can perform close to long-context LLMs. It further shows, contrary to the finding of a contemporary paper, that retrieval helps improves the LLM performance regardless of the context-window size. As a result, their best-performing model, a retrieval-augmented Llama2-70B-32K (Llama 2 finetuned to 32K context window with positional embedding interpolation) outperforms ChatGPT-3.5 on 5 of the 7 tasks.

**Strengths:**

- This paper investigates a widely-interested problem, long-context modeling, and compares two most popular solutions, long-context LLMs and retrieval-augmented LLMs. Its results and insights are hence of considerable interest to the research community.

- The paper is clearly written, and the experiments are relatively extensive. In particular, it achieved positive results showing retrieval augmentation and long-context LLMs can be complementary to each other, contradicting negative results from an earlier work. Furthermore, it experiments with rather capable SoTA LLMs such as Llama2-70B, instead of much smaller "toy" LLMs, making their results more convincing and useful.

**Weaknesses:**

- While this paper presents a useful empirical study, it does not introduce any new technique. Combining retrieval augmentation and long-context LLM is a straightforward idea, and the two being complementary to each other, though encouraging, is not very surprising.

- If I understand correctly (I find this part of the writing unclear, see Question 1 below and correct me if my assumption is wrong), the retrieval corpus is the collection of the context documents from each evaluation dataset (or maybe even each example). This may be a key reason why retrieval is more helpful in this work. In reality, for most of the tasks, it is impossible to assume to have perfectly relevant retrieval corpora (i.e. gold contexts from the test set), and the helpfulness of retrieval would dramatically decrease if the retrieval corpus mismatches the evaluation data.
As a result, the method considered in this paper is more tailored for one type of task, essentially reading comprehension of long documents, because the assumption that gold contexts, albeit long, are provided at test time. It would be more interesting to also explore the "open" settings where the gold context is not necessarily given.

**Questions:**

1. What's your retrieval corpus? Do you use the same corpus for all tasks? Or do you use the collection of gold contexts for each dataset as the retrieval corpus for that task? (If so, do you build a single index using contexts from all splits including the training and test set?) Or, do you build a separate index for each example using the chunks from the provided gold context for that example?
Sorry if I missed it, but I wasn't able to find any details on the retrieval corpora you used in your experiments.

2. It seems the evaluation tasks are all "closed-type", meaning the gold context is given at inference time. Is this the case? If so, how would retrieval be further helpful if the gold context is already provided? Is it because for certain tasks, the provided gold context can be too long and the relevant parts may have been truncated?

3. In Table 4, it is interesting that using the same top 5 retrieved chunks, DRAGON works better with 32k LLM while Contriever works better with 4k LLM. Any insights into what caused this? Have you tried reversing the order of the retrieved chunks?

---

> ### Author Response · Authors · 2023-11-23
> **Rebuttal by Authors**
>
> Thank you so much for your review. We will address your comments and questions in the following.
>
> Weakness:
>
> > “While this paper presents a useful empirical study, it does not introduce any new technique. Combining retrieval augmentation and long-context LLM is a straightforward idea, and the two being complementary to each other, though encouraging, is not very surprising.”
>   - We agree this paper does not introduce any new techniques, and the conclusion is “not very surprising”. However, we believe it is an important, timely and very useful contribution to the field. It investigates two important trends in LLMs i.e., retrieval-augmented generation (RAG) and long context LLMs. The obtained results are significant; for example, our best model, retrieval-augmented LLaMA2-70B with 32K context window, outperforms GPT-3.5-turbo-16k and Davinci003 in terms of average score on seven long context tasks including question answering and query-based summarization (see the updated Table 3 in the updated draft). The conclusion and insights are very useful for the practitioners; for example, RAG and long context LLMs should be combined to obtain the best results. We update the abstract of the paper to clarify the contribution of this work.
>
> > “If I understand correctly (I find this part of the writing unclear, see Question 1 below and correct me if my assumption is wrong), the retrieval corpus is the collection of the context documents from each evaluation dataset (or maybe even each example). This may be a key reason why retrieval is more helpful in this work. In reality, for most of the tasks, it is impossible to assume to have perfectly relevant retrieval corpora (i.e. gold contexts from the test set), and the helpfulness of retrieval would dramatically decrease if the retrieval corpus mismatches the evaluation data. As a result, the method considered in this paper is more tailored for one type of task, essentially reading comprehension of long documents, because the assumption that gold contexts, albeit long, are provided at test time. It would be more interesting to also explore the "open" settings where the gold context is not necessarily given.”
>   - Thanks for raising the question. Yes, the long context QA and summarization tasks are mostly similar to reading comprehension of long documents. Note that doing QA and summarization on customers' own documents (can be long for a lot of cases) is one of the major applications of long context LLM e.g., [1, 2]. The retrieval-augmentation certainly works for open-domain QA (e.g., retrieval corpus can be wikipedia). However, the current long context LLM (even up to 128K context length) can’t work in such “open” settings without retrieval.
>
>
> [1]  Anthropic. Introducing 100k context windows. 2023
>
> [2] OpenAI. Function calling and other API updates (longer context). 2-23
>
> Questions:
>
> > 1. “What's your retrieval corpus?”
>   - The retrieval corpus for each task/dataset is the provided single or multiple documents in that dataset. There is no training set for each task, as all the seven tasks are designed for zero-shot evaluation w/o any task-specific finetuning.
>
> > 2. “It seems the evaluation tasks are all "closed-type", meaning the gold context is given at inference time. Is this the case? If so, how would retrieval be further helpful if the gold context is already provided? Is it because for certain tasks, the provided gold context can be too long and the relevant parts may have been truncated?”
>   - Thanks for raising the question. Yes, the long context tasks are “closed-type”, but the given single or multiple gold documents are quite long e.g. NarrativeQA has average document length as 84k, and maximum document length as multiple hundred of thousand tokens. In many real world applications, a document can have million tokens e.g., detailed technical manuals. In these cases, retrieval is helpful, as i) the current long context LLM won’t be able to handle, ii) even it is able to handle, it is very expensive in terms of computation, iii) the sparsity provided by good retriever can improve accuracy than dense self-attention mechanism on long sequence, as suggestion by our retrieval-augmented Llama2-70B-32k outperforms its  Llama2-70B-32k baseline.
>
> > 3. “In Table 4, it is interesting that using the same top 5 retrieved chunks, DRAGON works better with 32k LLM while Contriever works better with 4k LLM. Any insights into what caused this? Have you tried reversing the order of the retrieved chunks?”
>   - Thanks for the question. We want to clarify that we did not intend to argue which retriever is better but rather that state-of-the-art retrievers can help boost the performance. As the differences are quite minor, we don’t have any clear insights for this. We tried the reversed order but we did not observe any improvements.

---

### Official Review · Reviewer_B6CQ · 2023-11-04

**Soundness:** 2 fair
**Presentation:** 3 good
**Contribution:** 3 good
**Rating:** 8
**Confidence:** 4

**Summary:**

This research paper compares two methods of handling context in large language models (LLMs); extending the context window and retrieval augmentation. The main contributions of this study are as follows;

1. The researchers conducted experiments using two state-of-the-art LLMs, a proprietary 43B GPT and LLaMA2 70B on seven tasks related to long context question answering and summarization.
2. The results demonstrate that retrieval significantly improves the performance of both 4K) and long (16K/32K) context LLMs. Surprisingly LLMs, with a context size of 4K, when combined with retrieval perform close to LLMs with a context size of 16K without incurring additional overhead.
3. It is also shown that extending the context window and retrieval augmentation complement each other. The performing model, LLaMA2 70B 32K enhanced by retrieval outperforms ChatGPT 3.5.

To summarize this paper provides insights into the combination of retrieval techniques and long context strategies for constructing improved language models (LLMs). The key findings indicate that retrieval is beneficial regardless of the context window size and extending the context window alongside retrieval augmentation offers reinforcing solutions, than competing ones.

**Strengths:**

This research study aims to examine and compare two approaches, for improving Large Language Models (LLMs); extended context utilization, and retrieval mechanisms. The authors utilize two LLMs with 43 billion and 70 billion parameters, which allow for better integration of context. Through experimentation, they assess the performance of these models across challenging datasets related to long context Question Answering and summarization.

The research provides insights demonstrating that retrieval mechanisms offer performance benefits regardless of the context length. Additionally, the study highlights how these two techniques complement each other. The paper is skillfully written, presenting an explanation of its motivation, background, experimental methodology, and analytical framework. The introduction effectively places the research in the context of work on long-context understanding, efficient attention mechanisms, and retrieval augmentation. This establishes a foundation for the study.

The authors thoughtfully compare their findings with research, including benchmark models like ChatGPT. This offers a view of the progress made in this field. Furthermore, the paper raises questions about comparing context understanding and retrieval as strategies for enhancing LLMs encouraging further exploration in this area. The extensive original experiments conducted contribute significantly to our understanding of these methods, within the community.

Overall the results of the study are highly significant, for the research community as they shed light on the effects of combining retrieval and long context.

**Weaknesses:**

Most studies primarily focus on improving models during inference with attention given to examining the impact of training methodologies.
There is a lack of analysis when it comes to integrating context and retrieval techniques like determining the ideal number of retrieved passages.
Experiments have not yet been conducted on contexts, which span 64,000 tokens and this could pose additional challenges.
There is a need, for evaluations against advanced methods in long context modeling and further benchmarking is necessary.
The experiments section would be more informative if it provided information about characteristics and task formats.
The related work section should clearly highlight the contributions of this study compared to work like LongBench.
There is a discussion regarding the implications and limitations associated with the techniques being investigated.
It's important to note that previous research has explored the concepts of context and retrieval. This study should acknowledge and build upon existing literature.
There are opportunities to explore approaches in combining retrieval and context that merit investigation.
Further validation is required to assess the suitability of these techniques, for contexts.

**Questions:**

The paper mentions another study called LongBench that reached conclusions regarding the impact of retrieval, on context models. It would be beneficial if the authors could further analyze the differences that led to these opposing observations.

Have you tried training the retriever and LLM from start to finish of just adding augmentation during inference? How does the performance compare in cases?

When retrieving passages how do you determine the optimal number to use? Does performance level off. Decline with many passages?

For context models like those with 64K or 128K tokens do you believe retrieval would still provide benefits?. Does its value decrease after a point?

Could you provide analysis and examples that demonstrate the phenomenon you hypothesize as "lost in the middle"?

Are there any approaches besides using long context and retrieval augmentation that you could compare to? For example memory or hierarchical attention.

Although the paper focuses on QA and summarization it would be interesting to see if similar conclusions apply to tasks.

Have you considered examining the impacts and limitations of building larger LLMs? A brief discussion, on this topic would be valuable.

---

> ### Author Response · Authors · 2023-11-23
> **Rebuttal by Authors (2/2)**
>
> > 7. “Have you tried training the retriever and LLM from start to finish of just adding augmentation during inference? How does the performance compare in cases?”
>   - Thank you for raising the question. We just add augmentation during inference in such retrieval-augmented generation (RAG) setting. Training the retriever and LLM in an end-to-end manner is an interesting direction for future exploration.
>
> > 8. “When retrieving passages how do you determine the optimal number to use? Does performance level off. Decline with many passages?”
>   - This is a good question. We conducted an ablation study using Llama-70B in Table 5 of our submission. We find that feeding top-5 retrieved passages to LLM averagely gives better performance than top-10 and top-20. Note that, although top-10 or top-20 passages have higher recall of relevant information, they also contain more irrelevant information than top-5, and could distract LLM from generating the right answer.
>
> > 9. “For context models like those with 64K or 128K tokens do you believe retrieval would still provide benefits?. Does its value decrease after a point?”
>   - This is a good question. We believe retrieval would certainly provide benefits, if the document length is longer than 64K or 128K tokens. Indeed, there are a lot of real-world documents having millions of tokens, where the retrieval-augmentation really shines. Another benefit of retrieval is about saving computation by utilizing an efficient retriever to select relevant context, rather than the expensive self-attention mechanism in LLM.
>
> > 10. “Could you provide analysis and examples that demonstrate the phenomenon you hypothesize as "lost in the middle?”
>   - Thanks for your suggestion. We conduct the “lost-in-the-middle” study as in Liu et al., (2023) for Llama2-70B-4k and Llama2-70B-32k. See Figure 1 in the updated draft. We confirm that the phenomenon also exists in Llama2 with different context lengths. In particular, the comparison of the curves from Llama2-70B-4k and Llama2-70B-32k suggests that the long context model has better accuracy for incorporating retrieved context.
>
> > 11. “Although the paper focuses on QA and summarization it would be interesting to see if similar conclusions apply to tasks.”
>   -  Thanks for your suggestion. We conduct few-shot experiment on Trec and SAMsum and the results are as follow:
>
> |                                | Trec   | SAMSum |
> |:-------------:               |:-------:|:-------------:|
> | GPT3.5-turbo-16k   |    68  |    41.7        |
> | Llama2-70b-32k     |    73   |    46.48      |
> | Llama2-70b-32k-ret |  76    |    47.31      |
>
> Our best model Llama2-70B-32k-ret outperforms its non-retrieval Llama2-70B-32k baseline as well as GPT-3.5-turbo-16k by a large margin. It again confirms the benefits of using retrieval together with long context models. We have included this part in Section 4.5
>
>
> > 12. “Have you considered examining the impacts and limitations of building larger LLMs? A brief discussion, on this topic would be valuable.”
>   - For larger LLM (e.g., > 100B), the benefit of RAG can be more significant in terms of saving computation, as larger LLMs require much more computation for long context input than smaller LLMs. On the other hand, larger LLM do have stronger capability to incorporate retrieved context, thus adding more retrieved context e.g., top-10 or top-20 might be better than top-5.

---

> ### Author Response · Authors · 2023-11-23
> **Rebuttal by Authors (1/2)**
>
> Many thanks for your detailed review & comments; they are really helpful to improve the quality of our paper. We will respond to your comments in the following.
>
> > 1. “There is a lack of analysis when it comes to integrating context and retrieval techniques like determining the ideal number of retrieved passages.”
>   - Thank you for the comment. In Table 5 of the submitted draft, we did a comparison with different numbers of retrieved passages from top-5, top-10 to top-20. From the results, there is no optimal predetermined number that works consistently better. It would be interesting to dynamically determine the optimal number of retrieved passages. We leave it for future study.
>
> > 2. “Experiments have not yet been conducted on contexts, which span 64,000 tokens and this could pose additional challenges.”
>   - Thank you for the valuable comment. Extending the context window to 64k and even longer would be very interesting. One of the major challenges is that training 64k context window LLM with 70B parameters is really expensive. Due to the limit of resources, we leave it for future study.
>
> > 3. “There is a need, for evaluations against advanced methods in long context modeling and further benchmarking is necessary. .” and “For example memory or hierarchical attention.”
>   - We agree with your comment in general. However, making advanced methods (e.g. memory or hierarchical attention) works for existing pretrained large language models e.g. Llama2-70B, is itself non-trivial. We include it in the future direction section of our updated draft.
>
> > 4. “The experiments section would be more informative if it provided information about characteristics and task formats.”
>   - Thank you for the valuable comments. We revised Section 3.2 and added more information about the characteristics and task formats for each task.
>
> > 5. “The related work section should clearly highlight the contributions of this study compared to work like LongBench.”
>   - Thank you for the valuable comment. In related work section 2.4 of submission, we describe the difference and contribution of this study compared to the LongBench work. In the updated manuscript, to highlight the discussion, we move this discussion to section 2.1.
>
> > 6. “The paper mentions another study called LongBench that reached conclusions regarding the impact of retrieval on context models. It would be beneficial if the authors could further analyze the differences that led to these opposing observations.”
>   - The LongBench work (Bai et al. 2023) finds that retrieval is only helpful for Llama2-7B-chat-4k with 4K context window, but not helpful for long context model ChatGLM2-6B-32k.
>   - We have added the results using Llama2-7B in the following Table and updated draft. One can actually draw similar conclusions to Bai et al. (2023). We think the underlying reasons are: i) For Llama2-7B-chat-4k, its short context length is the bottleneck for long context tasks. Thus, retrieval-augmentation largely improves the results. ii) For Llama2-7B-chat-32k and ChatGLM2-6B-32k, the context length bottleneck has been mostly removed. However, their retrieval-augmented models have limited zero-shot capability of incorporating retrieved chunks of context, due to the smaller size. As a result, retrieval is not helpful for both Llama2-7B-32k and ChatGLM2-6B-32k, which is different from large LLMs like Llama2-70B-32k. We also add this analysis and results into our updated manuscript.
>
> |        | Seq len             | Avg.          | QM | QASP | NQA | QLTY | MSQ | HQA | MFQA|
> | ------ | --------------- | ------- | -- | - | - | - | - | - | - |
> | Llama2-7B            | 4k   | 22.65  | 14.25  |	22.07 |	14.38  | 40.90 | 8.66  |	23.13  | 35.20|
> | Llama2-7B (+ret)  | 4k    | 26.04 | 16.45  |	22.97 |	18.18  | 43.25 | 14.68 | 26.62 |40.10 |
> | Llama2-7B            | 32k  | 28.20 | 16.09 | 23.66   |19.07   | 44.50 | 15.74 | 31.63 |46.71 |
> | Llama2-7B (+ret)  | 32k  | 27.63 | 17.11  | 	23.25  |19.12  | 43.70 | 15.67 | 29.55 | 45.03 |

---

### Official Review · Reviewer_1poR · 2023-11-04

**Soundness:** 2 fair
**Presentation:** 3 good
**Contribution:** 3 good
**Rating:** 6
**Confidence:** 3

**Summary:**

The paper investigates the comparison between retrieval-augmentation and long context extension for LLMs across various downstream tasks. It demonstrates that retrieval-augmentation can improve the performance of LLMs irrespective of context window size. They also reveal that shorter context window LLM with retrieval-augmentation can perform close to longer context LLM finetuned via positional interpolation, while taking much less computation.

**Strengths:**

1. The paper delves into the significant yet underexplored problem of retrieval-augmentation versus extended context windows, presenting a comprehensive study that spans a variety of popular long-context tasks to compare the two methods.
2. The findings that a "shorter context window LLM with simple retrieval-augmentation at inference can perform on par with a longer context LLM fine-tuned through positional interpolation," and that "retrieval can enhance the performance of LLMs regardless of their context window size," offer valuable insights for future research endeavors.

**Weaknesses:**

1. The authors attribute the superior performance of retrieval-augmented long context LLMs (e.g., 16K and 32K) over the 4K context LLM in a top-5 setting to the 'lost-in-the-middle' phenomenon. However, since the 'lost-in-the-middle' conclusions have not been widely recorganized, conducting an ablation study would be instrumental in supporting this hypothesis.
2. The authors use LLM performance metrics to infer the efficacy of different retrieval modules (Table 4). However, it is questionable whether a retrieval system with better LLM performance necessarily correlates with enhanced retrieval capability. The authors should incorporate some form of automated evaluation or a case study on retrieval performance to substantiate this.

**Questions:**

1. The OpenAI GPT series also offers variants with different context lengths (e.g., GPT-3.5-turbo, GPT-3.5-turbo-16k). Including their results in Table 2 and examining whether the conclusions still hold would be beneficial.
2. Regarding the results presented in Table 2, it is unclear how the best retriever was determined. Is a better-performing retriever module always associated with improved LLM performance?

---

> ### Author Response · Authors · 2023-11-23
> **Rebuttal by Authors**
>
> Many thanks for your comments and feedback. We will address your comments in the following.
>
> > 1. “The authors attribute the superior performance of retrieval-augmented long context LLMs (e.g., 16K and 32K) over the 4K context LLM in a top-5 setting to the 'lost-in-the-middle' phenomenon. However, since the 'lost-in-the-middle' conclusions have not been widely recorganized, conducting an ablation study would be instrumental in supporting this hypothesis.”
>   - Thank you for the valuable suggestion. We conduct the “lost-in-the-middle” study as in (Liu et al., 2023) for Llama2-70B-4k and Llama2-70B-32k. See Figure 1 in the updated draft. We confirm that the phenomenon also exists in Llama2 with different context lengths. In particular, the comparison of the curves from Llama2-70B-4k and Llama2-70B-32k suggests that the long context model has better accuracy for incorporating top-5 retrieved context.
>
>
> > 2. “The authors use LLM performance metrics to infer the efficacy of different retrieval modules (Table 4). However, it is questionable whether a retrieval system with better LLM performance necessarily correlates with enhanced retrieval capability. The authors should incorporate some form of automated evaluation or a case study on retrieval performance to substantiate this.”
>   - Thank you for the comment. We don’t intend to claim that one retriever is better than another based on the downstream LLM performance. Indeed, the difference of average accuracy of using Dragon (35.73 for Llama2-70B-4k), Contriever (36.02), and OpenAI-embedding (35.79) are relatively small compared to baseline Llama2-70B-4k w/o retrieval (31.61). So, the takeaway is that retrieval-augmented generation (RAG) with state-of-the-art retrievers can boost the accuracy of  baseline w/o retrieval. We further clarify this in our updated draft.
>
>
> > 3. “The OpenAI GPT series also offers variants with different context lengths (e.g., GPT-3.5-turbo, GPT-3.5-turbo-16k). Including their results in Table 2 and examining whether the conclusions still hold would be beneficial.”
>   - Many thanks for your valuable suggestions. In our updated draft, we provide the results of GPT-3.5-turbo (4k context length), GPT-3.5-turbo-16k in Table 3. We add the retrieval-augmentation results of GPT-3.5-turbo and  GPT-3.5-turbo-16k below for MSQ, HQA, MFQA for and also in Table 3
>
> |                                   | Avg | MSQ | HQA | MFQA|
> |--- | - |:-------------:|:-------------:|:-------------:|
> |GPT3.5-turbo-4k	| 37.08 | 21.23 | 40.86	| 49.16 |
> |GPT3.5-turbo-4k-ret	| 41.15 | 24.41 | 49.54 | 49.50 |
> |GPT3.5-turbo-16k	| 43.60 | 26.90 | 51.60 | 52.30 |
> |GPT3.5-turbo-16k-ret | 43.27	| 30.40 | 46.60 | 52.80 |
> |Llam2-70B-32k-ret     | 44.51	| 26.72 | 53.89 | 52.91 |
>
> For GPT3.5-turbo-4k, retrieval significantly improves the performance . For GPT3.5-turbo-16k, the average scores for retrieval and non-retrieval scores are close to each other which are both lower than our Llam2-70B-32k-ret results. Also note that GPT3.5-turbo-16k is a blackbox API, we don’t know how it is implemented, the model size as well as any preprocessing steps implemented. We add this part to our updated draft.
>
>
> > 4. “Regarding the results presented in Table 2, it is unclear how the best retriever was determined. Is a better-performing retriever module always associated with improved LLM performance?.”
>   - Thank you for the question. There is no guarantee that a better-performing retriever (in terms of precision/recall based evaluation metric) could always provide improved LLM performance. In Table 2, we simply pick the retriever based on the best LLM performance. Note that, the differences of LLM’s average performances associated with different SOTA retrievers, e.g. Dragon, Contriver and openai-embedding are relatively small as shown in Table 4.

---

### Meta-Review · Area_Chair_5RKR · 2023-12-12

**Metareview:**

There are two possible approaches to consider when handling longer sequences, extending an LMs context length or retrieving relevant pieces of information and use those with a short context LM.
The paper studies this exact tradeoff. Their findings suggest that retrieval significantly enhances the performance of both short short and long (32k) context LLMs, with the 4K LLMs augmented with retrieval achieving comparable results to 16K LLMs, and the top-performing retrieval based model surpasses GPT-3.5-turbo-16k and Davinci003 in downstream tasks.

While the paper's idea and execution seems straightforward, the reviewers generally liked the paper.
The strengths include the insightful demonstration of how retrieval mechanisms can enhance LLMs regardless of their context window size, the effective demonstration of the benefits of retrieval for long context models, and the clear motivation and extensive experimentation on a range of tasks.

Reviewers also raised some concerns including the need for more experiments (e.g., exploration of the similar lost-in-the-middle hypothesis (1poR)), a lack of analysis on the integration of context and retrieval techniques and the need for broader benchmarking (B6CQ), the absence of novel techniques and limitations in the experimental design (MoDw), and narrow focus (xWjb). It looks like some of the concerns have been largely addressed during the rebuttal.

**Justification For Why Not Higher Score:**

Please see the weaknesses sections in the metareview. Given these weaknesses, I don't feel comfortable recommending a stronger score.

**Justification For Why Not Lower Score:**

The reviewers all liked the paper.

---

### Decision · Program_Chairs · 2024-01-16

Accept (poster)